# UNBIASED SCALABLE SOFTMAX OPTIMIZATION

## ABSTRACT

Recent neural network and language models have begun to rely on softmax distributions with an extremely large number of categories. In this context calculating the softmax normalizing constant is prohibitively expensive. This has spurred a growing literature of efficiently computable but biased estimates of the softmax. In this paper we present the first two *unbiased* algorithms for maximizing the softmax likelihood whose work per iteration is independent of the number of classes and datapoints (and does not require extra work at the end of each epoch). We compare our unbiased methods' empirical performance to the state-of-the-art on seven real world datasets, where they comprehensively outperform all competitors.

## 1 INTRODUCTION

Under the softmax model[1] the probability that a random variable $y$ takes on the label $\ell \in \{1, ..., K\}$, is given by

$$p(y = \ell | x; W) = \frac{e^{x^\top w_\ell}}{\sum_{k=1}^{K} e^{x^\top w_k}}, \tag{1}$$

where $x \in \mathbb{R}^D$ is the covariate, $w_k \in \mathbb{R}^D$ is the vector of parameters for the $k$-th class, and $W = [w_1, w_2, ..., w_K] \in \mathbb{R}^{D \times K}$ is the parameter matrix. Given a dataset of $N$ label-covariate pairs $\mathcal{D} = \{(y_i, x_i)\}_{i=1}^{N}$, the ridge-regularized maximum log-likelihood problem is given by

$$L(W) = \sum_{i=1}^{N} x_i^\top w_{y_i} - \log(\sum_{k=1}^{K} e^{x_i^\top w_k}) - \frac{\mu}{2} \|W\|_2^2, \tag{2}$$

where $\|W\|_2$ denotes the Frobenius norm.

This paper focusses on how to maximize (2) when $N, K, D$ are all large. Having large $N, K, D$ is increasingly common in modern applications such as natural language processing and recommendation systems, where $N, K, D$ can each be on the order of millions or billions (Partalas et al., 2015; Chelba et al., 2013; Bhatia et al.).

A natural approach to maximizing $L(W)$ with large $N, K, D$ is to use Stochastic Gradient Descent (SGD), sampling a mini-batch of datapoints each iteration. However if $K, D$ are large then the $O(KD)$ cost of calculating the normalizing sum $\sum_{k=1}^{K} e^{x_i^\top w_k}$ in the stochastic gradients can still be prohibitively expensive. Several approximations that avoid calculating the normalizing sum have been proposed to address this difficulty. These include tree-structured methods (Bengio et al., 2003; Daume III et al., 2016; Grave et al., 2016), sampling methods (Bengio & Senécal, 2008; Mnih & Teh, 2012; Joshi et al., 2017) and self-normalization (Andreas & Klein, 2015). Alternative models such as the spherical family of losses (de Brébisson & Vincent, 2015; Vincent et al., 2015) that do not require normalization have been proposed to sidestep the issue entirely (Martins & Astudillo, 2016). Krishnapuram et al. (2005) avoid calculating the sum using a maximization-majorization approach based on lower-bounding the eigenvalues of the Hessian matrix. All[2] of these approximations are computationally tractable for large $N, K, D$, but are unsatisfactory in that they are biased and do not converge to the optimal $W^* = \text{argmax } L(W)$.

---

[1]Also known as the multinomial logit model.

[2]The method of Krishnapuram et al. (2005) does converge to the optimal MLE, but has $O(ND)$ runtime per iteration which is not feasible for large $N, D$.

Recently Raman et al. (2016) managed to recast (2) as a double-sum over $N$ and $K$. This formulation is amenable to SGD that samples both a datapoint and class each iteration, reducing the per iteration cost to $O(D)$. The problem is that vanilla SGD when applied to this formulation is unstable, in that the gradients suffer from high variance and are susceptible to computational overflow. Raman et al. (2016) deal with this instability by occasionally calculating the normalizing sum for all datapoints at a cost of $O(NKD)$. Although this achieves stability, its high cost nullifies the benefit of the cheap $O(D)$ per iteration cost.

The goal of this paper is to develop robust SGD algorithms for optimizing double-sum formulations of the softmax likelihood. We develop two such algorithms. The first is a new SGD method called U-max, which is guaranteed to have bounded gradients and converge to the optimal solution of (2) for all sufficiently small learning rates. The second is an implementation of Implicit SGD, a stochastic gradient method that is known to be more stable than vanilla SGD and yet has similar convergence properties (Toulis et al., 2016). We show that the Implicit SGD updates for the double-sum formulation can be efficiently computed and has a bounded step size, guaranteeing its stability.

We compare the performance of U-max and Implicit SGD to the (biased) state-of-the-art methods for maximizing the softmax likelihood which cost $O(D)$ per iteration. Both U-max and Implicit SGD outperform all other methods. Implicit SGD has the best performance with an average log-loss 4.29 times lower than the previous state-of-the-art.

In summary, our contributions in this paper are that we:

1. Provide a simple derivation of the softmax double-sum formulation and identify why vanilla SGD is unstable when applied to this formulation (Section 2).

2. Propose the U-max algorithm to stabilize the SGD updates and prove its convergence (Section 3.1).

3. Derive an efficient Implicit SGD implementation, analyze its runtime and bound its step size (Section 3.2).

4. Conduct experiments showing that both U-max and Implicit SGD outperform the previous state-of-the-art, with Implicit SGD having the best performance (Section 4).

## 2 CONVEX DOUBLE-SUM FORMULATION

### 2.1 DERIVATION OF DOUBLE-SUM

In order to have an SGD method that samples both datapoints and classes each iteration, we need to represent (2) as a double-sum over datapoints and classes. We begin by rewriting (2) in a more convenient form,

$$L(W) = \sum_{i=1}^{N} -\log(1 + \sum_{k \neq y_i} e^{x_i^\top (w_k - w_{y_i})}) - \frac{\mu}{2}\|W\|_2^2. \tag{3}$$

The key to converting (3) into its double-sum representation is to express the negative logarithm using its convex conjugate:

$$-\log(a) = \max_{v<0}\{av - (-\log(-v) - 1)\} = \max_{u}\{-u - \exp(-u)a + 1\} \tag{4}$$

where $u = -\log(-v)$ and the optimal value of $u$ is $u^*(a) = \log(a)$. Applying (4) to each of the logarithmic terms in (3) yields

$$L(W) = \sum_{i=1}^{N} \max_{u_i \in \mathbb{R}}\{-u_i - e^{-u_i}(1 + \sum_{k \neq y_i} e^{x_i^\top (w_k - w_{y_i})}) + 1\} - \frac{\mu}{2}\|W\|_2^2$$

$$= -\min_{u \geq 0}\{f(u, W)\} + N,$$

where

$$f(u, W) = \sum_{i=1}^{N} \sum_{k \neq y_i} \frac{u_i + e^{-u_i}}{K-1} + e^{x_i^\top (w_k - w_{y_i}) - u_i} + \frac{\mu}{2}\|W\|_2^2 \tag{5}$$

is our double-sum representation that we seek to minimize and the optimal solution for $u_i$ is $u_i^*(W) = \log(1 + \sum_{k \neq y_i} e^{x_i^\top (w_k - w_{y_i})}) \geq 0$. Clearly $f$ is a jointly convex function in $u$ and $W$. In Appendix A we prove that the optimal value of $u$ and $W$ is contained in a compact convex set and that $f$ is strongly convex within this set. Thus performing projected-SGD on $f$ is guaranteed to converge to a unique optimum with a convergence rate of $O(1/T)$ where $T$ is the number of iterations (Lacoste-Julien et al., 2012).

## 2.2 INSTABILITY OF VANILLA SGD

The challenge in optimizing $f$ using SGD is that it can have problematically large magnitude gradients. Observe that $f = \mathbb{E}_{ik}[f_{ik}]$ where $i \sim \text{unif}(\{1, ..., N\})$, $k \sim \text{unif}(\{1, ..., K\} - \{y_i\})$ and

$$f_{ik}(u, W) = N \left( u_i + e^{-u_i} + (K-1)e^{x_i^\top (w_k - w_{y_i}) - u_i} \right) + \frac{\mu}{2}(\beta_{y_i} \|w_{y_i}\|_2^2 + \beta_k \|w_k\|_2^2), \quad (6)$$

where $\beta_j = \frac{N}{n_j + (N - n_j)(K-1)}$ is the inverse of the probability of class $j$ being sampled either through $i$ or $k$, and $n_j = |\{i : y_i = j, i = 1, ..., N\}|$. The corresponding stochastic gradient is:

$$\nabla_{w_k} f_{ik}(u, W) = N(K-1)e^{x_i^\top (w_k - w_{y_i}) - u_i} x_i + \mu \beta_k w_k$$

$$\nabla_{w_{y_i}} f_{ik}(u, W) = -N(K-1)e^{x_i^\top (w_k - w_{y_i}) - u_i} x_i + \mu \beta_{y_i} w_{y_i}$$

$$\nabla_{w_{j \notin \{k, y_i\}}} f_{ik}(u, W) = 0$$

$$\nabla_{u_i} f_{ik}(u, W) = -N(K-1)e^{x_i^\top (w_k - w_{y_i}) - u_i} + N(1 - e^{-u_i}) \quad (7)$$

If $u_i$ equals its optimal value $u_i^*(W) = \log(1 + \sum_{k \neq y_i} e^{x_i^\top (w_k - w_{y_i})})$ then $e^{x_i^\top (w_k - w_{y_i}) - u_i} \leq 1$ and the magnitude of the $N(K-1)$ terms in the stochastic gradient are bounded by $N(K-1)\|x_i\|_2$. However if $u_i \ll x_i^\top (w_k - w_{y_i})$, then $e^{x_i^\top (w_k - w_{y_i}) - u_i} \gg 1$ and the magnitude of the gradients can become extremely large.

Extremely large gradients lead to two major problems: (a) the gradients may computationally overflow floating-point precision and cause the algorithm to crash, (b) they result in the stochastic gradient having high variance, which leads to slow convergence[3]. In Section 4 we show that these problems occur in practice and make vanilla SGD both an unreliable and inefficient method[4].

The sampled softmax optimizers in the literature (Bengio & Senécal, 2008; Mnih & Teh, 2012; Joshi et al., 2017) do not have the issue of large magnitude gradients. Their gradients are bounded by $N(K-1)\|x_i\|_2$ due to their approximations to $u_i^*(W)$ always being greater than $x_i^\top (w_k - w_{y_i})$. For example, in one-vs-each (Titsias, 2016), $u_i^*(W)$ is approximated by $\log(1 + e^{x_i^\top (w_k - w_{y_i})}) > x_i^\top (w_k - w_{y_i})$. However, as they only approximate $u_i^*(W)$ they cannot converge to the optimal $W^*$.

The goal of this paper is to design reliable and efficient SGD algorithms for optimizing the double-sum formulation $f(u, W)$ in (5). We propose two such methods: U-max (Section 3.1) and an implementation of Implicit SGD (Section 3.2). But before we introduce these methods we should establish that $f$ is a good choice for the double-sum formulation.

## 2.3 CHOICE OF DOUBLE-SUM FORMULATION

The double-sum in (5) is different to that of Raman et al. (2016). Their formulation can be derived by applying the convex conjugate substitution to (2) instead of (3). The resulting equations are $L(W) = \min_{\bar{u}} \left\{ \frac{1}{N} \sum_{i=1}^N \frac{1}{K-1} \sum_{k \neq y_i} \bar{f}_{ik}(\bar{u}, W) \right\} + N$ where

$$\bar{f}_{ik}(\bar{u}, W) = N \left( \bar{u}_i - x_i^\top w_{y_i} + e^{x_i^\top w_{y_i} - \bar{u}_i} + (K-1)e^{x_i^\top w_k - \bar{u}_i} \right) + \frac{\mu}{2}(\beta_{y_i} \|w_{y_i}\|_2^2 + \beta_k \|w_k\|_2^2)$$

$$(8)$$

---

[3]The convergence rate of SGD is inversely proportional to the second moment of its gradients (Lacoste-Julien et al., 2012).

[4]The same problems arise if we approach optimizing (3) via stochastic composition optimization (Wang et al., 2016). As is shown in Appendix B, stochastic composition optimization yields near-identical expressions for the stochastic gradients in (7) and has the same stability issues.

and the optimal solution for $\bar{u}_i$ is $\bar{u}_i^*(W^*) = \log(\sum_{k=1}^K e^{x_i^\top w_k^*})$.

Although both double-sum formulations can be used as a basis for SGD, our formulation tends to have smaller magnitude stochastic gradients and hence faster convergence. To see this, note that typically $x_i^\top w_{y_i} = \mathrm{argmax}_k\{x_i^\top w_k\}$ and so the $\bar{u}_i$, $x_i^\top w_{y_i}$ and $e^{x_i^\top w_{y_i} - \bar{u}_i}$ terms in (8) are of the greatest magnitude. Although at optimality these terms should roughly cancel, this will not be the case during the early stages of optimization, leading to stochastic gradients of large magnitude. In contrast the function $f_{ik}$ in (6) only has $x_i^\top w_{y_i}$ appearing as a negative exponent, and so if $x_i^\top w_{y_i}$ is large then the magnitude of the stochastic gradients will be small. In Section 4 we present numerical results confirming that our double-sum formulation leads to faster convergence.

## 3 STABLE SGD METHODS

### 3.1 U-MAX METHOD

As explained in Section 2.2, vanilla SGD has large gradients when $u_i \ll x_i^\top(w_k - w_{y_i})$. This can only occur when $u_i$ is less than its optimum value for the current $W$, since $u_i^*(W) = \log(1 + \sum_{j \neq y_i} e^{x_i^\top(w_k - w_{y_i})}) \geq x_i^\top(w_k - w_{y_i})$. A simple remedy is to set $u_i = \log(1 + e^{x_i^\top(w_k - w_{y_i})})$ whenever $u_i \ll x_i^\top(w_k - w_{y_i})$. Since $\log(1 + e^{x_i^\top(w_k - w_{y_i})}) > x_i^\top(w_k - w_{y_i})$ this guarantees that $u_i > x_i^\top(w_k - w_{y_i})$ and so the gradients are bounded. It also brings $u_i$ closer[5] to its optimal value for the current $W$ and thereby decreases the the objective $f(u, W)$.

This is exactly the mechanism behind the U-max algorithm — see Algorithm 1 in Appendix C for its pseudocode. U-max is the same as vanilla SGD except for two modifications: (a) $u_i$ is set equal to $\log(1 + e^{x_i^\top(w_k - w_{y_i})})$ whenever $u_i \leq \log(1 + e^{x_i^\top(w_k - w_{y_i})}) - \delta$ for some threshold $\delta > 0$, (b) $u_i$ is projected onto $[0, B_u]$, and $W$ onto $\{W : \|W\|_2 \leq B_W\}$, where $B_u$ and $B_W$ are set so that the optimal $u_i^* \in [0, B_u]$ and the optimal $W^*$ satisfies $\|W^*\|_2 \leq B_W$. See Appendix A for more details on how to set $B_u$ and $B_W$.

**Theorem 1.** *Let $B_f = \max_{\|W\|_2^2 \leq B_W^2, 0 \leq u \leq B_u}, \max_{ik} \|\nabla f_{ik}(u, W)\|_2$. Suppose a learning rate $\eta_t \leq \delta^2/(4B_f^2)$, then U-max with threshold $\delta$ converges to the optimum of (2), and the rate is at least as fast as SGD with same learning rate, in expectation.*

*Proof.* The proof is provided in Appendix D. $\qquad\square$

U-max directly resolves the problem of extremely large gradients. Modification (a) ensures that $\delta \geq x_i^\top(w_k - w_{y_i}) - u_i$ (otherwise $u_i$ would be increased to $\log(1 + e^{x_i^\top(w_k - w_{y_i})})$) and so the magnitude of the U-max gradients are bounded above by $N(K - 1)e^\delta\|x_i\|_2$.

In U-max there is a trade-off between the gradient magnitude and learning rate that is controlled by $\delta$. For Theorem 1 to apply we require that the learning rate $\eta_t \leq \delta^2/(4B_f^2)$. A small $\delta$ yields small magnitude gradients, which makes convergence fast, but necessitates a small $\eta_t$, which makes convergence slow.

### 3.2 IMPLICIT SGD

Another method that solves the large gradient problem is Implicit SGD[6] (Bertsekas, 2011; Toulis et al., 2016). Implicit SGD uses the update equation

$$\theta^{(t+1)} = \theta^{(t)} - \eta_t \nabla f(\theta^{(t+1)}, \xi_t), \tag{9}$$

where $\theta^{(t)}$ is the value of the $t^{th}$ iterate, $f$ is the function we seek to minimize and $\xi_t$ is a random variable controlling the stochastic gradient such that $\nabla f(\theta) = \mathbb{E}_{\xi_t}[\nabla f(\theta, \xi_t)]$. The update (9) differs from vanilla SGD in that $\theta^{(t+1)}$ appears on both the left and right side of the equation,

---

[5]Since $u_i < x_i^\top(w_k - w_{y_i}) < \log(1 + e^{x_i^\top(w_k - w_{y_i})}) < \log(1 + \sum_{j \neq y_i} e^{x_i^\top(w_k - w_{y_i})}) = u_i^*(W)$.
[6]Also known to as an "incremental proximal algorithm" (Bertsekas, 2011).

whereas in vanilla SGD it appears only on the left side. In our case $\theta = (u, W)$ and $\xi_t = (i_t, k_t)$ with $\nabla f(\theta^{(t+1)}, \xi_t) = \nabla f_{i_t, k_t}(u^{(t+1)}, W^{(t+1)})$.

Although Implicit SGD has similar convergence rates to vanilla SGD, it has other properties that can make it preferable over vanilla SGD. It is known to be more robust to the learning rate (Toulis et al., 2016), which important since a good value for the learning rate is never known a priori. Another property, which is of particular interest to our problem, is that it has smaller step sizes.

**Proposition 1.** *Consider applying Implicit SGD to optimizing $f(\theta) = \mathbb{E}_\xi[f(\theta, \xi)]$ where $f(\theta, \xi)$ is $m$-strongly convex for all $\xi$. Then*

$$\|\nabla f(\theta^{(t+1)}, \xi_t)\|_2 \le \|\nabla f(\theta^{(t)}, \xi_t)\|_2 - m\|\theta^{(t+1)} - \theta^{(t)}\|_2$$

*and so the Implicit SGD step size is smaller than that of vanilla SGD.*

*Proof.* The proof is provided in Appendix E. □

The bound in Proposition 1 can be tightened for our particular problem. Unlike vanilla SGD whose step size magnitude is *exponential* in $x_i^\top(w_k - w_{y_i}) - u_i$, as shown in (7), for Implicit SGD the step size is asymptotically *linear* in $x_i^\top(w_k - w_{y_i}) - u_i$. This effectively guarantees that Implicit SGD cannot suffer from computational overflow.

**Proposition 2.** *Consider the Implicit SGD algorithm where in each iteration only one datapoint $i$ and one class $k \ne y_i$ is sampled and there is no ridge regularization. The magnitude of its step size in $w$ is $O(x_i^\top(w_k - w_{y_i}) - u_i)$.*

*Proof.* The proof is provided in Appendix F.2. □

The difficulty in applying Implicit SGD is that in each iteration one has to compute a solution to (9). The tractability of this procedure is problem dependent. We show that computing a solution to (9) is indeed tractable for the problem considered in this paper. The details of these mechanisms are laid out in full in Appendix F.

**Proposition 3.** *Consider the Implicit SGD algorithm where in each iteration $n$ datapoints and $m$ classes are sampled. Then the Implicit SGD update $\theta^{(t+1)}$ can be computed to within $\epsilon$ accuracy in runtime $O(n(n + m)(D + n\log(\epsilon^{-1})))$.*

*Proof.* The proof is provided in Appendix F.3. □

In Proposition 3 the $\log(\epsilon^{-1})$ factor comes from applying a first order method to solve the strongly convex Implicit SGD update equation. It may be the case that performing this optimization is more expensive than computing the $x_i^\top w_k$ inner products, and so each iteration of Implicit SGD may be significantly slower than that of vanilla SGD or U-max. However, in the special case of $n = m = 1$ we can use the bisection method to give an explicit upper bound on the optimization cost.

**Proposition 4.** *Consider the Implicit SGD algorithm with learning rate $\eta$ where in each iteration only one datapoint $i$ and one class $k \ne y_i$ is sampled and there is no ridge regularization. Then the Implicit SGD iterate $\theta^{(t+1)}$ can be computed to within $\epsilon$ accuracy with only two $D$-dimensional vector inner products and at most $\log_2(\epsilon^{-1}) + \log_2(|x_i^\top(w_k - w_{y_i}) - u_i| + 2\eta N\|x_i\|_2^2 + \log(K - 1))$ bisection method function evaluations.*

*Proof.* The proof is provided in Appendix F.1 □

For any reasonably large dimension $D$, the cost of the two $D$-dimensional vector inner products will outweigh the cost of the bisection, and Implicit SGD will have roughly the same speed per iteration as vanilla SGD or U-max.

In summary, Implicit SGD is robust to the learning rate, does not have overflow issues and its updates can be computed in roughly the same time as vanilla SGD.

Table 1: Datasets with a summary of their properties. Where the number of classes, dimension or number of examples has been altered, the original value is displayed in brackets.

| DATASET | CLASSES | DIMENSION | EXAMPLES |
|---|---|---|---|
| MNIST | 10 | 780 | 60,000 |
| Bibtex | 147 (159) | 1,836 | 4,880 |
| Delicious | 350 (983) | 500 | 12,920 |
| Eurlex | 838 (3,993) | 5,000 | 15,539 |
| AmazonCat-13K | 2,709 (2,919) | 10,000 (203,882) | 100,000 (1,186,239) |
| Wiki10 | 4,021 (30,938) | 10,000 (101,938) | 14,146 |
| Wiki-small | 18,207 (28,955) | 10,000 (2,085,164) | 90,737 (342,664) |

## 4 EXPERIMENTS

Two sets of experiments were conducted to assess the performance of the proposed methods. The first compares U-max and Implicit SGD to the state-of-the-art over seven real world datasets. The second investigates the difference in performance between the two double-sum formulations discussed in Section 2.3. We begin by specifying the experimental setup and then move onto the results.

### 4.1 EXPERIMENTAL SETUP

**Data.** We used the MNIST, Bibtex, Delicious, Eurlex, AmazonCat-13K, Wiki10, and Wiki-small datasets[7], the properties of which are summarized in Table 1. Most of the datasets are multi-label and, as is standard practice (Titsias, 2016), we took the first label as being the true label and discarded the remaining labels. To make the computation more manageable, we truncated the number of features to be at most 10,000 and the training and test size to be at most 100,000. If, as a result of the dimension truncation, a datapoint had no non-zero features then it was discarded. The features of each dataset were normalized to have unit $L_2$ norm. All of the datasets were pre-separated into training and test sets. We only focus on the performance on the algorithms on the training set, as the goal in this paper is to investigate how best to optimize the softmax likelihood, which is given over the training set.

**Algorithms.** We compared our algorithms to the state-of-the-art methods for optimizing the softmax which have runtime $O(D)$ per iteration[8]. The competitors include Noise Contrastive Estimation (NCE) (Mnih & Teh, 2012), Importance Sampling (IS) (Bengio & Senécal, 2008) and One-Vs-Each (OVE) (Titsias, 2016). Note that these methods are all biased and will not converge to the optimal softmax MLE, but something close to it. For these algorithms we set $n = 100, m = 5$, which are standard settings[9]. For Implicit SGD we chose to implement the version in Proposition 4 which has $n = 1, m = 1$. Likewise for U-max we set $n = 1, m = 1$ and the threshold parameter $\delta = 1$. The ridge regularization parameter $\mu$ was set to zero for all algorithms.

**Epochs and losses.** Each algorithm is run for 50 epochs on each dataset. The learning rate is decreased by a factor of 0.9 each epoch. Both the prediction error and log-loss (2) are recorded at the end of 10 evenly spaced epochs over the 50 epochs.

**Learning rate.** The magnitude of the gradient differs in each algorithm, due to either under- or over-estimating the log-sum derivative from (2). To set a reasonable learning rate for each algorithm on

---

[7]All of the datasets were downloaded from `http://manikvarma.org/downloads/XC/XMLRepository.html`, except Wiki-small which was obtained from `http://lshtc.iit.demokritos.gr/`.

[8]Raman et al. (2016) have runtime $O(NKD)$ per epoch, which is equivalent to $O(KD)$ per iteration. This is a factor of $K$ slower than the methods we compare against.

[9]We also experimented setting $n = 1, m = 1$ in these methods and there was virtually no difference except the runtime was slower. For example, in Appendix G we plot the performance of NCE with $n = 1, m = 1$ and $n = 100, m = 5$ applied to the Eurlex dataset for different learning rates and there is very little difference between the two.

Table 2: Tuned initial learning rates for each algorithm on each dataset. The learning rate in $10^{0,\pm1,\pm2,\pm3}$ with the lowest log-loss after 50 epochs using only 10% of the data is displayed. Vanilla SGD applied to AmazonCat, Wiki10 and Wiki-small suffered from overflow with a learning rate of $10^{-3}$, but was stable with smaller learning rates (the largest learning rate for which it was stable is displayed).

| DATASET | OVE | NCE | IS | Vanilla | U-max | Implicit |
|---|---|---|---|---|---|---|
| MNIST | $10^1$ | $10^1$ | $10^1$ | $10^{-2}$ | $10^1$ | $10^{-1}$ |
| Bibtex | $10^2$ | $10^2$ | $10^2$ | $10^{-2}$ | $10^{-1}$ | $10^1$ |
| Delicious | $10^1$ | $10^3$ | $10^3$ | $10^{-3}$ | $10^{-2}$ | $10^{-2}$ |
| Eurlex | $10^{-1}$ | $10^2$ | $10^2$ | $10^{-3}$ | $10^{-1}$ | $10^1$ |
| AmazonCat | $10^1$ | $10^3$ | $10^3$ | $10^{-5}$ | $10^{-2}$ | $10^{-3}$ |
| Wiki10 | $10^{-2}$ | $10^3$ | $10^2$ | $10^{-4}$ | $10^{-2}$ | $10^0$ |
| Wiki-small | $10^3$ | $10^3$ | $10^3$ | $10^{-4}$ | $10^{-3}$ | $10^{-3}$ |

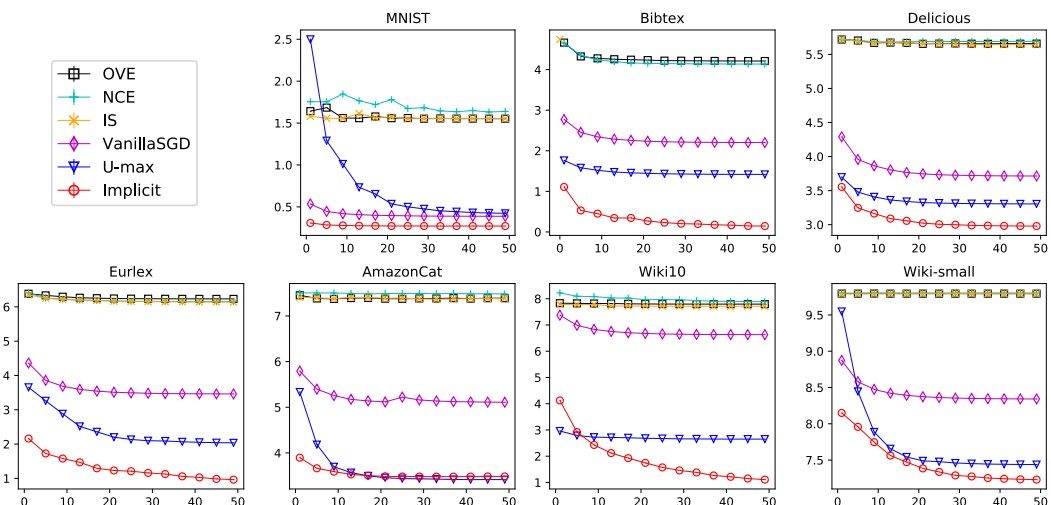

Figure 1: The x-axis is the number of epochs and the y-axis is the log-loss from (2) calculated at the current value of $W$.

each dataset, we ran them on 10% of the training data with initial learning rates $\eta = 10^{0,\pm1,\pm2,\pm3}$. The learning rate with the best performance after 50 epochs is then used when the algorithm is applied to the full dataset. The tuned learning rates are presented in Table 2. Note that vanilla SGD requires a very small learning rate, otherwise it suffered from overflow.

## 4.2 RESULTS

**Comparison to state-of-the-art.** Plots of the performance of the algorithms on each dataset are displayed in Figure 1 with the relative performance compared to Implicit SGD given in Table 3. The Implicit SGD method has the best performance on virtually all datasets. Not only does it converge faster in the first few epochs, it also converges to the optimal MLE (unlike the biased methods that prematurely plateau). On average after 50 epochs, Implicit SGD's log-loss is a factor of 4.29 lower than the previous state-of-the-art. The U-max algorithm also outperforms the previous state-of-the-art on most datasets. U-max performs better than Implicit SGD on AmazonCat, although in general Implicit SGD has superior performance. Vanilla SGD's performance is better than the previous state-of-the-art but worse than U-max and Implicit SGD. The difference in performance between vanilla SGD and U-max can largely be explained by vanilla SGD requiring a smaller learning rate to avoid computational overflow.

Table 3: Relative log-loss. The values for each dataset are normalized by dividing by the corresponding Implicit SGD log-loss. The lowest log-loss for each dataset is in bold.

| DATASET | OVE | NCE | IS | VANILLA-SGD | U-MAX. | IMPLICIT SGD |
|---|---|---|---|---|---|---|
| MNIST | 5.73 | 6.05 | 5.74 | 1.43 | 1.56 | **1.00** |
| Bibtex | 29.03 | 28.52 | 32.71 | 15.18 | 9.77 | **1.00** |
| Delicious | 1.90 | 1.91 | 1.89 | 1.25 | 1.11 | **1.00** |
| Eurlex | 6.47 | 6.39 | 6.38 | 3.59 | 2.11 | **1.00** |
| AmazonCat | 2.12 | 2.15 | 2.12 | 1.47 | **0.98** | 1.00 |
| Wiki10 | 7.04 | 7.13 | 6.97 | 5.99 | 2.39 | **1.00** |
| Wiki-small | 1.02 | 1.36 | 1.35 | 1.15 | 1.03 | **1.00** |
| Average | 7.62 | 7.64 | 8.17 | 4.29 | 2.71 | **1.00** |

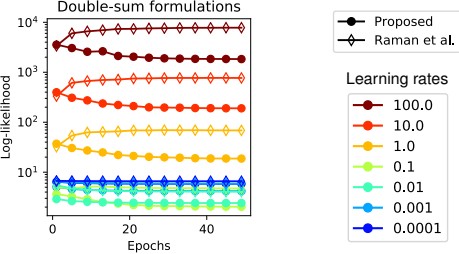

Figure 2: Log-loss of U-max on Eurlex for different learning rates with our proposed double-sum formulation and that of Raman et al. (2016).

The sensitivity of each method to the initial learning rate can be seen in Appendix G, where the results of running each method on the Eurlex dataset with learning rates $\eta = 10^{0,\pm1,\pm2,\pm3}$ is presented. The results are consistent with those in Figure 1, with Implicit SGD having the best performance for most learning rate settings. For learning rates $\eta = 10^{3,4}$ the U-max log-loss is extremely large. This can be explained by Theorem 1, which does not guarantee convergence for U-max if the learning rate is too high.

**Comparison of double-sum formulations.** Figure 2 illustrates the performance on the Eurlex dataset of U-max using the proposed double-sum in (6) compared to U-max using the double-sum of Raman et al. (2016) in (8). The proposed double-sum clearly outperforms for all[10] learning rates $\eta = 10^{0,\pm1,\pm2,-3,-4}$, with its $50^{th}$-epoch log-loss being 3.08 times lower on average. This supports the argument from Section 2.3 that SGD methods applied to the proposed double-sum have smaller magnitude gradients and converge faster.

## 5 CONCLUSION

In this paper we have presented the U-max and Implicit SGD algorithms for optimizing the softmax likelihood. These are the *first* algorithms that require only $O(D)$ computation per iteration (without extra work at the end of each epoch) that converge to the optimal softmax MLE. Implicit SGD can be efficiently implemented and clearly out-performs the previous state-of-the-art on seven real world datasets. The result is a new method that enables optimizing the softmax for extremely large number of samples and classes.

So far Implicit SGD has only been applied to the simple softmax, but could also be applied to any neural network where the final layer is the softmax. Applying Implicit SGD to word2vec type models, which can be viewed as softmaxes where both $x$ and $w$ are parameters to be fit, might be particularly fruitful.

---

[10]The learning rates $\eta = 10^{3,4}$ are not displayed in the Figure 2 for visualization purposes. It had similar behavior as $\eta = 10^2$.

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

## A    PROOF OF VARIABLE BOUNDS AND STRONG CONVEXITY

We first establish that the optimal values of $u$ and $W$ are bounded. Next, we show that within these bounds the objective is strongly convex and its gradients are bounded.

**Lemma 1** ((Raman et al., 2016)). *The optimal value of $W$ is bounded as $\|W^*\|_2^2 \leq B_W^2$ where $B_W^2 = \frac{2}{\mu} N \log(K)$.*

*Proof.*

$$-N \log(K) = L(0) \leq L(W^*) \leq -\frac{\mu}{2} \|W^*\|_2^2$$

Rearranging gives the desired result.    $\square$

**Lemma 2.** *The optimal value of $u_i$ is bounded as $u_i^* \leq B_u$ where $B_u = \log(1 + (K-1)e^{2B_x B_w})$ and $B_x = \max_i\{\|x_i\|_2\}$*

*Proof.*

$$u_i^* = \log(1 + \sum_{k \neq y_i} e^{x_i^\top (w_k - w_{y_i})})$$

$$\leq \log(1 + \sum_{k \neq y_i} e^{\|x_i\|_2(\|w_k\|_2 + \|w_{y_i}\|_2)})$$

$$\leq \log(1 + \sum_{k \neq y_i} e^{2B_x B_w})$$

$$= \log(1 + (K-1)e^{2B_x B_w})$$

$\square$

**Lemma 3.** *If $\|W\|_2^2 \leq B_W^2$ and $u_i \leq B_u$ then $f(u, W)$ is strongly convex with convexity constant greater than or equal to $\min\{\exp(-B_u), \mu\}$.*

*Proof.* Let us rewrite $f$ as

$$f(u, W) = \sum_{i=1}^N u_i + e^{-u_i} + \sum_{k \neq y_i} e^{x_i^\top (w_k - w_{y_i}) - u_i} + \frac{\mu}{2} \|W\|_2^2$$

$$= \sum_{i=1}^N a_i^\top \theta + e^{-u_i} + \sum_{k \neq y_i} e^{b_{ik}^\top \theta} + \frac{\mu}{2} \|W\|_2^2.$$

where $\theta = (u^\top, w_1^\top, ..., w_k^\top) \in \mathbb{R}^{N+KD}$ with $a_i$ and $b_{ik}$ being appropriately defined. The Hessian of $f$ is

$$\nabla^2 f(\theta) = \sum_{i=1}^N e^{-u_i} e_i e_i^\top + \sum_{k \neq y_i} e^{b_{ik}^\top \theta} b_{ik} b_{ik}^\top + \mu \cdot diag\{0_N, 1_{KD}\}$$

where $e_i$ is the $i^{th}$ canonical basis vector, $0_N$ is an $N$-dimensional vector of zeros and $1_{KD}$ is a $KD$-dimensional vector of ones. It follows that

$$\nabla^2 f(\theta) \succeq I \cdot \min\{\min_{0 \leq u \leq B_u} \{e^{-u_i}\}, \mu\}$$

$$= I \cdot \min\{\exp(-B_u), \mu\}$$

$$\succeq 0.$$

$\square$

**Lemma 4.** *If $\|W\|_2^2 \leq B_W^2$ and $u_i \leq B_u$ then the 2-norm of both the gradient of $f$ and each stochastic gradient $f_{ik}$ are bounded by*

$$B_f = N \max\{1, e^{B_u} - 1\} + 2(Ne^{B_u} B_x + \mu \max_k\{\beta_k\} B_W).$$

*Proof.* By Jensen's inequality

$$\max_{\|W\|_2^2 \le B_W^2, 0 \le u \le B_u} \|\nabla f(u,W)\|_2 = \max_{\|W\|_2^2 \le B_W^2, 0 \le u \le B_u} \|\nabla \mathbb{E}_{ik} f_{ik}(u,W)\|_2$$

$$\le \max_{\|W\|_2^2 \le B_W^2, 0 \le u \le B_u} \mathbb{E}_{ik} \|\nabla f_{ik}(u,W)\|_2$$

$$\le \max_{\|W\|_2^2 \le B_W^2, 0 \le u \le B_u} \max_{ik} \|\nabla f_{ik}(u,W)\|_2.$$

Using the results from Lemmas 1 and 2 and the definition of $f_{ik}$ from (6),

$$\|\nabla_{u_i} f_{ik}(u,W)\|_2 = \|N \left(1 - e^{-u_i} - (K-1)e^{x_i^\top (w_k - w_{y_i}) - u_i}\right)\|_2$$

$$= N|1 - e^{-u_i}(1 + (K-1)e^{x_i^\top (w_k - w_{y_i})})|$$

$$\le N \max\{1, (1 + (K-1)e^{\|x_i\|_2 (\|w_k\|_2 + \|w_{y_i}\|_2)}) - 1\}$$

$$\le N \max\{1, e^{B_u} - 1\}$$

and for $j$ indexing either the sampled class $k \ne y_i$ or the true label $y_i$,

$$\|\nabla_{w_j} f_{ik}(u,W)\|_2 = \| \pm N(K-1)e^{x_i^\top (w_k - w_{y_i}) - u_i} x_i + \mu \beta_j w_j \|_2$$

$$\le N(K-1)e^{\|x_i\|_2 (\|w_k\|_2 + \|w_{y_i}\|_2)} \|x_i\|_2 + \mu \beta_j \|w_j\|_2$$

$$\le Ne^{B_u} B_x + \mu \max_k \{\beta_k\} B_W.$$

Letting

$$B_f = N \max\{1, e^{B_u} - 1\} + 2(Ne^{B_u} B_x + \mu \max_k \{\beta_k\} B_W)$$

we have

$$\|\nabla f_{ik}(u,W)\|_2 \le \|\nabla_{u_i} f_{ik}(u,W)\|_2 + \|\nabla_{w_k} f_{ik}(u,W)\|_2 + \|\nabla_{w_{y_i}} f_{ik}(u,W)\|_2 = B_f.$$

In conclusion:

$$\max_{\|W\|_2^2 \le B_W^2, 0 \le u \le B_u} \|\nabla f(u,W)\|_2 \le \max_{\|W\|_2^2 \le B_W^2, u_i \le B_u,} \max_{ik} \|\nabla f_{ik}(u,W)\|_2 \le B_f.$$

$\square$

## B  STOCHASTIC COMPOSITION OPTIMIZATION

We can write the equation for $L(W)$ from (3) as (where we have set $\mu = 0$ for notational simplicity),

$$L(W) = -\sum_{i=1}^N \log(1 + \sum_{k \ne y_i} e^{x_i^\top (w_k - w_{y_i})})$$

$$= \mathbb{E}_i[h_i(\mathbb{E}_k[g_k(W)])]$$

where $i \sim unif(\{1,...,N\})$, $k \sim unif(\{1,...,K\})$, $h_i(v) \in \mathbb{R}$, $g_k(W) \in \mathbb{R}^N$ and

$$h_i(v) = -N \log(1 + e_i^\top v)$$

$$[g_k(W)]_i = \begin{cases} Ke^{x_i^\top (w_k - w_{y_i})} & \text{if } k \ne y_i \\ 0 & \text{otherwise} \end{cases}.$$

Here $e_i^\top v = v_i \in \mathbb{R}$ is a variable that is explicitly kept track of with $v_i \approx \mathbb{E}_k[g_k(W)]_i = \sum_{k \ne y_i} e^{x_i^\top (w_k - w_{y_i})}$ (with exact equality in the limit as $t \to \infty$). Clearly $v_i$ in stochastic composition optimization has a similar role as $u_i$ has in our formulation for $f$ in (5).

If $i, k$ are sampled with $k \ne y_i$ in stochastic composition optimization then the updates are of the form (Wang et al., 2016)

$$w_{y_i} = w_{y_i} + \eta_t NK \frac{e^{x_i^\top (z_k - z_{y_i})}}{1 + v_i} x_i$$

$$w_k = w_k - \eta_t NK \frac{e^{x_i^\top (z_k - z_{y_i})}}{1 + v_i} x_i,$$

where $z_k$ is a smoothed value of $w_k$. These updates have the same numerical instability issues as vanilla SGD on $f$ in (5): it is possible that $\frac{e^{x_i^\top z_k}}{1 + v_i} \gg 1$ where ideally we should have $0 \le \frac{e^{x_i^\top z_k}}{1 + v_i} \le 1$.

## C  U-MAX PSEUDOCODE

---

**Algorithm 1:** U-max

**Input**  : Data $\mathcal{D} = \{(y_i, x_i) : y_i \in \{1, \ldots, K\}, x_i \in \mathbb{R}^d\}_{i=1}^N$, number of classes $K$, number of datapoints $N$, learning rate $\eta_t$, class sampling probability $\beta_k = \frac{N}{n_k + (N - n_k)(K-1)}$, threshold parameter $\delta > 0$, bound $B_W$ on $W$ such that $\|W\|_2 \leq B_W$ and bound $B_u$ on $u$ such that $u_i \leq B_u$ for $i = 1, \ldots, N$

**Output:** $W$

1  Initialize
2  **for** $k = 1$ **to** $K$ **do**
3  $\quad$ $w_k \leftarrow 0$
4  **end**
5  **for** $i = 1$ **to** $N$ **do**
6  $\quad$ $u_i \leftarrow \log(K)$
7  **end**

8  Run SGD
9  **for** $t = 1$ **to** $T$ **do**
10 $\quad$ Sample indices
11 $\quad$ $i \sim unif(\{1, \ldots, N\})$
12 $\quad$ $k \sim unif(\{1, \ldots, K\} - \{y_i\})$

13 $\quad$ Increase $u_i$
14 $\quad$ **if** $u_i < \log(1 + e^{x_i^\top (w_k - w_{y_i})}) - \delta$ **then**
15 $\quad\quad$ $u_i \leftarrow \log(1 + e^{x_i^\top (w_k - w_{y_i})})$

16 $\quad$ SGD step
17 $\quad$ $w_k \leftarrow w_k - \eta_t [N(K-1)e^{x_i^\top (w_k - w_{y_i}) - u_i} x_i + \mu \beta_k w_k]$
18 $\quad$ $w_{y_i} \leftarrow w_{y_i} - \eta_t [-N(K-1)e^{x_i^\top (w_k - w_{y_i}) - u_i} x_i + \mu \beta_{y_i} w_{y_i}]$
19 $\quad$ $u_i \leftarrow u_i - \eta_t [N(1 - e^{-u_i} - (K-1)e^{x_i^\top (w_k - w_{y_i}) - u_i})]$

20 $\quad$ Projection
21 $\quad$ $w_k \leftarrow w_k \cdot \min\{1, B_W / \|w_k\|_2\}$
22 $\quad$ $w_{y_i} \leftarrow w_{y_i} \cdot \min\{1, B_W / \|w_{y_i}\|_2\}$
23 $\quad$ $u_i \leftarrow \max\{0, \min\{B_u, u_i\}\}$
24 **end**

---

## D  PROOF OF CONVERGENCE OF U-MAX METHOD

In this section we will prove the claim made in Theorem 1, that U-max converges to the softmax optimum. Before proving the theorem, we will need a lemma.

**Lemma 5.** *For any $\delta > 0$, if $u_i \leq \log(1 + e^{x_i^\top (w_k - w_{y_i})}) - \delta$ then setting $u_i = \log(1 + e^{x_i^\top (w_k - w_{y_i})})$ decreases $f(u, W)$ by at least $\delta^2/2$.*

*Proof.* As in Lemma 3, let $\theta = (u^\top, w_1^\top, \ldots, w_k^\top) \in \mathbb{R}^{N+KD}$. Then setting $u_i = \log(1 + e^{x_i^\top (w_k - w_{y_i})})$ is equivalent to setting $\theta = \theta + \Delta e_i$ where $e_i$ is the $i^{th}$ canonical basis vector and $\Delta = \log(1 + e^{x_i^\top (w_k - w_{y_i})}) - u_i \geq \delta$. By a second order Taylor series expansion

$$f(\theta) - f(\theta + \Delta e_i) \geq \nabla f(\theta + \Delta e_i)^\top e_i \Delta + \frac{\Delta^2}{2} e_i^\top \nabla^2 f(\theta + \lambda \Delta e_i) e_i \qquad (10)$$

for some $\lambda \in [0, 1]$. Since the optimal value of $u_i$ for a given value of $W$ is $u_i^*(W) = \log(1 + \sum_{k \neq y_i} e^{x_i^\top (w_k - w_{y_i})}) \geq \log(1 + e^{x_i^\top (w_k - w_{y_i})})$, we must have $\nabla f(\theta + \Delta e_i)^\top e_i \leq 0$. From Lemma 3

we also know that

$$
\begin{aligned}
e_i^\top \nabla^2 f(\theta + \lambda \Delta e_i) e_i &= \exp(-(u_i + \lambda \Delta)) + \sum_{k \neq y_i} e^{x_i^\top (w_k - w_{y_i}) - (u_i + \lambda \Delta)} \\
&= \exp(-\lambda \Delta) e^{-u_i} \big(1 + \sum_{k \neq y_i} e^{x_i^\top (w_k - w_{y_i})}\big) \\
&= \exp(-\lambda \Delta) \exp(-(\log(1 + e^{x_i^\top (w_k - w_{y_i})}) - \Delta))\big(1 + \sum_{k \neq y_i} e^{x_i^\top (w_k - w_{y_i})}\big) \\
&\geq \exp(\Delta - \lambda \Delta) \\
&\geq \exp(\Delta - \Delta) \\
&= 1.
\end{aligned}
$$

Putting in bounds for the gradient and Hessian terms in (10),

$$
f(\theta) - f(\theta + \Delta e_i) \geq \frac{\Delta^2}{2} \geq \frac{\delta^2}{2}.
$$

$\square$

Now we are in a position to prove Theorem 1.

*Proof of Theorem 1.* Let $\theta^{(t)} = (u^{(t)}, W^{(t)}) \in \Theta$ denote the value of the $t^{th}$ iterate. Here $\Theta = \{\theta : \|W\|_2^2 \leq B_W^2, u_i \leq B_u\}$ is a convex set containing the optimal value of $f(\theta)$.

Let $\pi_i^{(\delta)}(\theta)$ denote the operation of setting $u_i = \log(1 + e^{x_i^\top (w_k - w_{y_i})})$ if $u_i \leq \log(1 + e^{x_i^\top (w_k - w_{y_i})}) - \delta$. If indices $i, k$ are sampled for the stochastic gradient and $u_i \leq \log(1 + e^{x_i^\top (w_k - w_{y_i})}) - \delta$, then the value of $f$ at the $t + 1^{st}$ iterate is bounded as

$$
\begin{aligned}
f(\theta^{(t+1)}) &= f(\pi_i(\theta^{(t)}) - \eta_t \nabla f_{ik}(\pi_i(\theta^{(t)}))) \\
&\leq f(\pi_i(\theta^{(t)})) + \max_{\theta \in \Theta} \|\eta_t \nabla f_{ik}(\pi_i(\theta))\|_2 \max_{\theta \in \Theta} \|\nabla f(\theta)\|_2 \\
&\leq f(\pi_i(\theta^{(t)})) + \eta_t B_f^2 \\
&\leq f(\theta^{(t)}) - \delta^2/2 + \eta_t B_f^2 \\
&\leq f(\theta^{(t)} - \eta_t \nabla f_{ik}(\theta^{(t)})) - \delta^2/2 + 2\eta_t B_f^2 \\
&\leq f(\theta^{(t)} - \eta_t \nabla f_{ik}(\theta^{(t)})),
\end{aligned}
$$

since $\eta_t \leq \delta^2/(4B_f^2)$ by assumption. Alternatively if $u_i \geq \log(1 + e^{x_i^\top (w_k - w_{y_i})}) - \delta$ then

$$
\begin{aligned}
f(\theta^{(t+1)}) &= f(\pi_i(\theta^{(t)}) - \eta_t \nabla f_{ik}(\pi_i(\theta^{(t)}))) \\
&= f(\theta^{(t)} - \eta_t \nabla f_{ik}(\theta^{(t)})).
\end{aligned}
$$

Either way $f(\theta^{(t+1)}) \leq f(\theta^{(t)} - \eta_t \nabla f_{ik}(\theta^{(t)}))$. Taking expectations with respect to $i, k$,

$$
\mathbb{E}_{ik}[f(\theta^{(t+1)})] \leq \mathbb{E}_{ik}[f(\theta^{(t)} - \eta_t \nabla f_{ik}(\theta^{(t)}))].
$$

Finally let $P$ denote the projection of $\theta$ onto $\Theta$. Since $\Theta$ is a convex set containing the optimum we have $f(P(\theta)) \leq f(\theta)$ for any $\theta$, and so

$$
\mathbb{E}_{ik}[f(P(\theta^{(t+1)}))] \leq \mathbb{E}_{ik}[f(\theta^{(t)} - \eta_t \nabla f_{ik}(\theta^{(t)}))],
$$

which shows that the rate of convergence in expectation of U-max is at least as fast as that of standard SGD.

$\square$

## E    PROOF OF GENERAL IMPLICIT SGD GRADIENT BOUND

*Proof of Theorem 2.* Let $f(\theta, \xi)$ be $m$-strongly convex for all $\xi$. The vanilla SGD step size is $\eta_t \|\nabla f(\theta^{(t)}, \xi_t)\|_2$ where $\eta_t$ is the learning rate for the $t^{th}$ iteration. The Implicit SGD step size is $\eta_t \|\nabla f(\theta^{(t+1)}, \xi_t)\|_2$ where $\theta^{(t+1)}$ satisfies $\theta^{(t+1)} = \theta^{(t)} - \eta_t \nabla f(\theta^{(t+1)}, \xi_t)$. Rearranging, $\nabla f(\theta^{(t+1)}, \xi_t) = (\theta^{(t)} - \theta^{(t+1)})/\eta_t$ and so it must be the case that $\nabla f(\theta^{(t+1)}, \xi_t)^\top (\theta^{(t)} - \theta^{(t+1)}) = \|\nabla f(\theta^{(t+1)}, \xi_t)\|_2 \|\theta^{(t)} - \theta^{(t+1)}\|_2$.

Our desired result follows:

$$
\begin{aligned}
\|\nabla f(\theta^{(t)}, \xi_t)\|_2 &\geq \frac{\nabla f(\theta^{(t)})^\top (\theta^{(t)} - \theta^{(t+1)})}{\|\theta^{(t)} - \theta^{(t+1)}\|_2} \\
&\geq \frac{\nabla f(\theta^{(t+1)})^\top (\theta^{(t)} - \theta^{(t+1)}) + m\|\theta^{(t)} - \theta^{(t+1)}\|_2^2}{\|\theta^{(t)} - \theta^{(t+1)}\|_2} \\
&= \frac{\|\nabla f(\theta^{(t+1)})\|_2 \|\theta^{(t)} - \theta^{(t+1)}\|_2 + m\|\theta^{(t)} - \theta^{(t+1)}\|_2^2}{\|\theta^{(t)} - \theta^{(t+1)}\|_2} \\
&= \|\nabla f(\theta^{(t+1)})\|_2 + m\|\theta^{(t)} - \theta^{(t+1)}\|_2
\end{aligned}
$$

where the first inequality is by Cauchy-Schwarz and the second inequality by strong convexity.

$\square$

## F    UPDATE EQUATIONS FOR IMPLICIT SGD

In this section we will derive the updates for Implicit SGD. We will first consider the simplest case where only one datapoint $(x_i, y_i)$ and a single class is sampled in each iteration with no regularizer. Then we will derive the more complicated update for when there are multiple datapoints and sampled classes with a regularizer.

### F.1    SINGLE DATAPOINT, SINGLE CLASS, NO REGULARIZER

Equation (6) for the stochastic gradient for a single datapoint and single class with $\mu = 0$ is

$$
f_{ik}(u, W) = N(u_i + e^{-u_i} + (K-1)e^{x_i^\top (w_k - w_{y_i}) - u_i}).
$$

The Implicit SGD update corresponds to finding the variables optimizing

$$
\min_{u, W} \left\{ 2\eta f_{ik}(u, W) + \|u - \tilde{u}\|_2^2 + \|W - \tilde{W}\|_2^2 \right\},
$$

where $\eta$ is the learning rate and the tilde refers to the value of the old iterate (Toulis et al., 2016, Eq. 6). Since $f_{ik}$ is only a function of $u_i, w_k, w_{y_i}$ the optimization reduces to

$$
\min_{u_i, w_k, w_{y_i}} \left\{ 2\eta f_{ik}(u_i, w_k, w_{y_i}) + (u_i - \tilde{u}_i)^2 + \|w_{y_i} - \tilde{w}_{y_i}\|_2^2 + \|w_k - \tilde{w}_k\|_2^2 \right\}
$$

$$
= \min_{u_i, w_k, w_{y_i}} \left\{ 2\eta N(u_i + e^{-u_i} + (K-1)e^{x_i^\top (w_k - w_{y_i}) - u_i}) \right.
$$

$$
\left. + (u_i - \tilde{u}_i)^2 + \|w_{y_i} - \tilde{w}_{y_i}\|_2^2 + \|w_k - \tilde{w}_k\|_2^2 \right\}.
$$

The optimal value of $w_k, w_{y_i}$ must deviate from the old value $\tilde{w}_k, \tilde{w}_{y_i}$ in the direction of $x_i$. Furthermore we can observe that the deviation of $w_k$ must be exactly opposite that of $w_{y_i}$, that is:

$$
w_{y_i} = \tilde{w}_{y_i} + a \frac{x_i}{2\|x_i\|_2^2}
$$

$$
w_k = \tilde{w}_k - a \frac{x_i}{2\|x_i\|_2^2} \tag{11}
$$

for some $a \geq 0$. The optimization problem reduces to

$$
\min_{u_i, a \geq 0} \left\{ 2\eta N(u_i + e^{-u_i} + (K-1)e^{x_i^\top (\tilde{w}_k - \tilde{w}_{y_i})} e^{-a - u_i}) + (u_i - \tilde{u}_i)^2 + a^2 \frac{1}{2\|x_i\|_2^2} \right\}. \tag{12}
$$

We'll approach this optimization problem by first solving for $a$ as a function of $u_i$ and then optimize over $u_i$. Once the optimal value of $u_i$ has been found, we can calculate the corresponding optimal value of $a$. Finally, substituting $a$ into (11) will give us our updated value of $W$.

**Solving for $a$**

We solve for $a$ by setting its derivative equal to zero in (12)

$$0 = \partial_a \left\{ 2\eta N(u_i + e^{-u_i} + (K-1)e^{x_i^\top(\tilde{w}_k - \tilde{w}_{y_i})}e^{-a-u_i}) + (u_i - \tilde{u}_i)^2 + a^2 \frac{1}{2\|x_i\|_2^2} \right\}$$

$$= -2\eta N(K-1)e^{x_i^\top(\tilde{w}_k - \tilde{w}_{y_i})-u_i}e^{-a} + a\frac{1}{\|x_i\|_2^2}$$

$$\Leftrightarrow ae^a = 2\eta N(K-1)\|x_i\|_2^2 e^{x_i^\top(\tilde{w}_k - \tilde{w}_{y_i})-u_i}. \tag{13}$$

The solution for $a$ can be written in terms of the principle branch of the Lambert W function $P$,

$$a(u_i) = P(2\eta N(K-1)\|x_i\|_2^2 e^{x_i^\top(\tilde{w}_k - \tilde{w}_{y_i})-u_i})$$

$$= P(e^{x_i^\top(\tilde{w}_k - \tilde{w}_{y_i})-u_i+\log(2\eta N(K-1)\|x_i\|_2^2)}). \tag{14}$$

Substituting the solution to $a(u_i)$ into (12), we now only need minimize over $u_i$:

$$\min_{u_i} \left\{ 2\eta N u_i + 2\eta N e^{-u_i} + 2\eta N(K-1)e^{x_i^\top(\tilde{w}_k - \tilde{w}_{y_i})}e^{-a(u_i)-u_i} + (u_i - \tilde{u}_i)^2 + a(u_i)^2\frac{1}{2\|x_i\|_2^2} \right\}$$

$$= \min_{u_i} \left\{ 2\eta N u_i + 2\eta N e^{-u_i} + a(u_i)\|x_i\|_2^{-2} + (u_i - \tilde{u}_i)^2 + a(u_i)^2\frac{1}{2\|x_i\|_2^2} \right\} \tag{15}$$

where we used the fact that $e^{-P(z)} = P(z)/z$. The derivative with respect to $u_i$ in (15) is

$$\partial_{u_i} \left\{ 2\eta N u_i + 2\eta N e^{-u_i} + a(u_i)\|x_i\|_2^{-2} + (u_i - \tilde{u}_i)^2 + a(u_i)^2\frac{1}{2\|x_i\|_2^2} \right\}$$

$$= 2\eta N - 2\eta N e^{-u_i} + \partial_{u_i}a(u_i)\|x_i\|_2^{-2} + 2(u_i - \tilde{u}_i) + 2a(u_i)\partial_{u_i}a(u_i)\frac{1}{2\|x_i\|_2^2}$$

$$= 2\eta N - 2\eta N e^{-u_i} - \frac{a(u_i)}{1 + a(u_i)}\|x_i\|_2^{-2} + 2(u_i - \tilde{u}_i) - \frac{a(u_i)^2}{(1 + a(u_i))\|x_i\|_2^2} \tag{16}$$

where to calculate $\partial_{u_i}a(u_i)$ we used the fact that $\partial_z P(z) = \frac{P(z)}{z(1+P(z))}$ and so

$$\partial_{u_i}a(u_i) = -\frac{a(u_i)}{e^{x_i^\top(\tilde{w}_k - \tilde{w}_{y_i})-u_i+\log(2\eta N(K-1)\|x_i\|_2^2)}(1 + a(u_i))}e^{x_i^\top(\tilde{w}_k - \tilde{w}_{y_i})-u_i+\log(2\eta N(K-1)\|x_i\|_2^2)}$$

$$= -\frac{a(u_i)}{1 + a(u_i)}.$$

**Bisection method for $u_i$**

We can solve for $u_i$ using the bisection method. Below we show how to calculate the initial lower and upper bounds of the bisection interval and prove that the size of the interval is bounded (which ensures fast convergence).

Start by calculating the derivative in (16) at $u_i = \tilde{u}_i$. If the derivative is negative then the optimal $u_i$ is lower bounded by $\tilde{u}_i$. An upper bound is provided by

$$u_i = \underset{u_i}{\operatorname{argmin}} \left\{ 2\eta N(u_i + e^{-u_i} + (K-1)e^{x_i^\top(\tilde{w}_k - \tilde{w}_{y_i})}e^{-a(u_i)-u_i}) + (u_i - \tilde{u}_i)^2 + \frac{a(u_i)^2}{2\|x_i\|_2^2} \right\}$$

$$\leq \underset{u_i}{\operatorname{argmin}} \left\{ 2\eta N(u_i + e^{-u_i} + (K-1)e^{x_i^\top(\tilde{w}_k - \tilde{w}_{y_i})}e^{-u_i}) + (u_i - \tilde{u}_i)^2 \right\}$$

$$\leq \underset{u_i}{\operatorname{argmin}} \left\{ 2\eta N(u_i + e^{-u_i} + (K-1)e^{x_i^\top(\tilde{w}_k - \tilde{w}_{y_i})}e^{-u_i}) \right\}$$

$$= \log(1 + (K-1)e^{x_i^\top(\tilde{w}_k - \tilde{w}_{y_i})}).$$

In the first inequality we set $a(u_i) = 0$, since by the envelop theorem the gradient of $u_i$ is monotonically increasing in $a$. In the second inequality we used the assumption that $u_i$ is lower bounded by $\tilde{u}_i$. Thus if the derivative in (16) is negative at $u_i = \tilde{u}_i$ then $\tilde{u}_i \le u_i \le \log(1 + (K-1)e^{x_i^\top(\tilde{w}_k - \tilde{w}_{y_i})})$. If $(K-1)e^{x_i^\top(\tilde{w}_k - \tilde{w}_{y_i})} \le 1$ then the size of the interval must be less than $\log(2)$, since $\tilde{u}_i \ge 0$. Otherwise the gap must be at most $\log(2(K-1)e^{x_i^\top(\tilde{w}_k - \tilde{w}_{y_i})}) - \tilde{u}_i = \log(2(K-1)) + x_i^\top(\tilde{w}_k - \tilde{w}_{y_i}) - \tilde{u}_i$. Either way, the gap is upper bounded by $\log(2(K-1)) + |x_i^\top(\tilde{w}_k - \tilde{w}_{y_i}) - \tilde{u}_i|$.

Now let us consider if the derivative in (16) is positive at $u_i = \tilde{u}_i$. Then $u_i$ is upper bounded by $\tilde{u}_i$. Denoting $a'$ as the optimal value of $a$, we can lower bound $u_i$ using (12)

$$
\begin{aligned}
u_i &= \operatorname*{argmin}_{u_i} \left\{ 2\eta N(u_i + e^{-u_i} + (K-1)e^{x_i^\top(\tilde{w}_k - \tilde{w}_{y_i})}e^{-a'-u_i}) + (u_i - \tilde{u}_i)^2 \right\} \\
&\ge \operatorname*{argmin}_{u_i} \left\{ u_i + e^{-u_i} + (K-1)e^{x_i^\top(\tilde{w}_k - \tilde{w}_{y_i})}e^{-a'-u_i} \right\} \\
&= \log(1 + (K-1)\exp(x_i^\top(\tilde{w}_k - \tilde{w}_{y_i}) - a')) \\
&\ge \log(K-1) + x_i^\top(\tilde{w}_k - \tilde{w}_{y_i}) - a'
\end{aligned}
\tag{17}
$$

where the first inequality comes dropping the $(u_i - \tilde{u}_i)^2$ term due to the assumption that $u_i < \tilde{u}_i$. Recall (13),

$$
a'e^{a'} = 2\eta N(K-1)\|x_i\|_2^2 e^{x_i^\top(\tilde{w}_k - \tilde{w}_{y_i}) - u_i}.
$$

The solution for $a'$ is strictly monotonically increasing as a function of the right side of the equation. Thus replacing the right side with an upper bound on its value results in an upper bound on $a'$. Substituting the bound for $u_i$,

$$
\begin{aligned}
a' &\le \min\{a : ae^a = 2\eta N(K-1)\|x_i\|_2^2 e^{x_i^\top(\tilde{w}_k - \tilde{w}_{y_i}) - (\log(K-1) + x_i^\top(\tilde{w}_k - \tilde{w}_{y_i}) - a)}\} \\
&= \min\{a : a = 2\eta N\|x_i\|_2^2\} \\
&= 2\eta N\|x_i\|_2^2.
\end{aligned}
\tag{18}
$$

Substituting this bound for $a'$ into (17) yields

$$
u_i \ge \log(K-1) + x_i^\top(\tilde{w}_k - \tilde{w}_{y_i}) - 2\eta N\|x_i\|_2^2.
$$

Thus if the derivative in (16) is positive at $u_i = \tilde{u}_i$ then $\log(K-1) + x_i^\top(\tilde{w}_k - \tilde{w}_{y_i}) - 2\eta N\|x_i\|_2^2 \le u_i \le \tilde{u}_i$. The gap between the upper and lower bound is $\tilde{u}_i - x_i^\top(\tilde{w}_k - \tilde{w}_{y_i}) + 2\eta N\|x_i\|_2^2 - \log(K-1)$.

In summary, for both cases of the sign of the derivative in (16) at $u_i = \tilde{u}_i$ we are able to calculate a lower and upper bound on the optimal value of $u_i$ such that the gap between the bounds is at most $|\tilde{u}_i - x_i^\top(\tilde{w}_k - \tilde{w}_{y_i})| + 2\eta N\|x_i\|_2^2 + \log(K-1)$. This allows us to perform the bisection method where for $\epsilon > 0$ level accuracy we require only $\log_2(\epsilon^{-1}) + \log_2(|\tilde{u}_i - x_i^\top(\tilde{w}_k - \tilde{w}_{y_i})| + 2\eta N\|x_i\|_2^2 + \log(K-1))$ function evaluations.

## F.2 Bound on step size

Here we will prove that the step size magnitude of Implicit SGD with a single datapoint and sampled class with respect to $w$ is bounded as $O(x_i^\top(\tilde{w}_k - \tilde{w}_{y_i}) - \tilde{u}_i)$. We will do so by considering the two cases $u_i' \ge \tilde{u}_i$ and $u_i' < \tilde{u}_i$ separately, where $u_i'$ denotes the optimal value of $u_i$ in the Implicit SGD update and $\tilde{u}_i$ is its value at the previous iterate.

**Case:** $u_i' \ge \tilde{u}_i$
Let $a'$ denote the optimal value of $a$ in the Implicit SGD update. From (14)

$$
\begin{aligned}
a' &= a(u_i') \\
&= P(e^{x_i^\top(\tilde{w}_k - \tilde{w}_{y_i}) - u_i' + \log(2\eta N(K-1)\|x_i\|_2^2)}) \\
&= P(e^{x_i^\top(\tilde{w}_k - \tilde{w}_{y_i}) - \tilde{u}_i + \log(2\eta N(K-1)\|x_i\|_2^2)}).
\end{aligned}
$$

Now using the fact that $P(z) = O(\log(z))$,

$$
\begin{aligned}
a' &= O(x_i^\top(\tilde{w}_k - \tilde{w}_{y_i}) - \tilde{u}_i + \log(2\eta N(K-1)\|x_i\|_2^2)) \\
&= O(x_i^\top(\tilde{w}_k - \tilde{w}_{y_i}) - \tilde{u}_i)
\end{aligned}
$$

**Case:** $u_i' < \tilde{u}_i$

If $u_i' < \tilde{u}_i$ then we can lower bound $a'$ from (18) as

$$a' \le 2\eta N \|x_i\|_2^2.$$

**Combining cases**

Putting together the two cases,

$$a' = O(\max\{x_i^\top(\tilde{w}_k - \tilde{w}_{y_i}) - \tilde{u}_i, \ 2\eta N \|x_i\|_2^2\})$$
$$= O(x_i^\top(\tilde{w}_k - \tilde{w}_{y_i}) - \tilde{u}_i).$$

The actual step size in $w$ is $\pm a \frac{x_i}{2\|x_i\|_2^2}$. Since $a$ is $O(x_i^\top(\tilde{w}_k - \tilde{w}_{y_i}) - \tilde{u}_i)$, the step size magnitude is also $O(x_i^\top(\tilde{w}_k - \tilde{w}_{y_i}) - \tilde{u}_i)$.

### F.3 MULTIPLE DATAPOINTS, MULTIPLE CLASSES

The Implicit SGD update when there are multiple datapoints, multiple classes, with a regularizer is similar to the singe datapoint, singe class, no regularizer case described above. However, there are a few significant differences. Firstly, we will require some pre-computation to find a low-dimensional representation of the $x$ values in each mini-batch. Secondly, we will integrate out $u_i$ for each datapoint (not $w_k$). And thirdly, since the dimensionality of the simplified optimization problem is large, we'll require first order or quasi-Newton methods to find the optimal solution.

#### F.3.1 DEFINING THE MINI-BATCH

The first step is to define our mini-batches of size $n$. We will do this by partitioning the datapoint indices into sets $S_1, ..., S_J$ with $S_j = \{j_\ell : \ell = 1, ..., n\}$ for $j = 1, ..., \lfloor N/n \rfloor$, $S_J = \{J_\ell : \ell = 1, ..., N \bmod n\}$, $S_i \cap S_j = \varnothing$ and $\cup_{j=1}^J S_j = \{1, ..., N\}$.

Next we define the set of classes $C_j$ which can be sampled for the $j^{th}$ mini-batch. The set $C_j$ is defined to be all sets of $m$ distinct classes that are not equal to any of the labels $y$ for points in the mini-batch, that is, $C_j = \{(k_1, ..., k_m) : k_i \in \{1, ..., K\}, \ k_i \ne k_\ell \forall \ell \in \{1, ..., m\} - \{i\}, \ k_i \ne y_\ell \forall \ell \in S_j\}$.

Now we can write down our objective from (5) in terms of an expectation of functions corresponding to our mini-batches:

$$f(u, W) = \mathbb{E}[f_{j,C}(u, W)]$$

where $j$ is sampled with probability $p_j = |S_j|/N$ and $C$ is sampled uniformly from $C_j$ and

$$f_{j,C}(u, W) = p_j^{-1} \sum_{i \in S_j} \left( u_i + e^{-u_i} + \sum_{k \in S_j - \{i\}} e^{x_i^\top(w_k - w_{y_i}) - u_i} + \frac{K-n}{m} \sum_{k \in C} e^{x_i^\top(w_k - w_{y_i}) - u_i} \right)$$
$$+ \frac{\mu}{2} \sum_{k \in C \cup S_j} \beta_k \|w_k\|_2^2.$$

The value of the regularizing constant $\beta_k$ is such that $\mathbb{E}[I[k \in C \cup S_j]\beta_k] = 1$, which requires that

$$\beta_k^{-1} = 1 - \frac{1}{J} \sum_{j=1}^J I[k \ne S_j](1 - \frac{m}{K - |S_j|}).$$

#### F.3.2 SIMPLIFYING THE IMPLICIT SGD UPDATE EQUATION

The Implicit SGD update corresponds to solving

$$\min_{u,W} \left\{ 2\eta f_{j,C}(u, W) + \|u - \tilde{u}\|_2^2 + \|W - \tilde{W}\|_2^2 \right\},$$

where $\eta$ is the learning rate and the tilde refers to the value of the old iterate (Toulis et al., 2016, Eq. 6). Since $f_{j,C}$ is only a function of $u_{S_j} = \{u_i : i \in S_j\}$ and $W_{j,C} = \{w_k : k \in S_j \cup C\}$ the optimization reduces to

$$\min_{u_{S_j}, W_{j,C}} \left\{ 2\eta f_{j,C}(u_{S_j}, W_{j,C}) + \|u_{S_j} - \tilde{u}_{S_j}\|_2^2 + \|W_{j,C} - \tilde{W}_{j,C}\|_2^2 \right\}.$$

The next step is to analytically minimize the $u_{S_j}$ terms. The optimization problem in (21) decomposes into a sum of separate optimization problems in $u_i$ for $i \in S_j$,

$$\min_{u_i} \left\{ 2\eta p_j^{-1}(u_i + e^{-u_i} d_i) + (u_i - \tilde{u}_i)^2 \right\}$$

where

$$d_i(W_{j,C}) = 1 + \sum_{k \in S_j - \{i\}} e^{x_i^\top(w_k - w_{y_i})} + \frac{K - n}{m} \sum_{k \in C} e^{x_i^\top(w_k - w_{y_i})}.$$

Setting the derivative of $u_i$ equal to zero yields the solution

$$u_i(W_{j,C}) = \tilde{u}_i - \eta p_j^{-1} + P(\eta p_j^{-1} d_i(W_{j,C}) \exp(\eta p_j^{-1} - \tilde{u}_i))$$

where $P$ is the principle branch of the Lambert W function. Substituting this solution into our optimization problem and simplifying yields

$$\min_{W_{j,C}} \left\{ \sum_{i \in S_j}(1 + P(\eta p_j^{-1} d_i(W_{j,C}) \exp(\eta p_j^{-1} - \tilde{u}_i)))^2 + \|W_{j,C} - \tilde{W}_{j,C}\|_2^2 + \frac{\mu}{2} \sum_{k \in C \cup S_j} \beta_k \|w_k\|_2^2 \right\},$$
(19)

where we have used the identity $e^{-P(z)} = P(z)/z$. We can decompose (19) into two parts by splitting $W_{j,C} = W_{j,C}^{\parallel} + W_{j,C}^{\perp}$, its components parallel and perpendicular to the span of $\{x_i : i \in S_j\}$ respectively. Since the leading term in (19) only depends on $W_{j,C}^{\parallel}$, the two resulting sub-problems are

$$\min_{W_{j,C}^{\parallel}} \left\{ \sum_{i \in S_j}(1 + P(\eta p_j^{-1} d_i(W_{j,C}^{\parallel}) \exp(\eta p_j^{-1} - \tilde{u}_i)))^2 + \|W_{j,C}^{\parallel} - \tilde{W}_{j,C}^{\parallel}\|_2^2 + \frac{\mu}{2} \sum_{k \in C \cup S_j} \beta_k \|w_k^{\parallel}\|_2^2 \right\},$$

$$\min_{W_{j,C}^{\perp}} \left\{ \|W_{j,C}^{\perp} - \tilde{W}_{j,C}^{\perp}\|_2^2 + \frac{\mu}{2} \sum_{k \in C \cup S_j} \beta_k \|w_k^{\perp}\|_2^2 \right\}$$
(20)

Let us focus on the perpendicular component first. Simple calculus yields the optimal value $w_k^{\perp} = \frac{1}{1 + \mu\beta_k/2}\tilde{w}_k^{\perp}$ for $k \in S_j \cup C$.

Moving onto the parallel component, let the span of $\{x_i : i \in S_j\}$ have an orthonormal basis[11] $V_j = (v_{j1}, ..., v_{jn}) \in \mathbb{R}^{D \times n}$ with $x_i = V_j b_i$ for some $b_i \in \mathbb{R}^n$. With this basis we can write $w_k^{\parallel} = \tilde{w}_k^{\parallel} + V_j a_k$ for $a_k \in \mathbb{R}^n$ which reduces the parallel component optimization problem to[12]

$$\min_{A_{j,C}} \left\{ \sum_{i \in S_j}(1 + P(z_{ijC}(A_{j,C})))^2 + \sum_{k \in S_j \cup C}(1 + \frac{\mu\beta_k}{2})\|a_k\|_2^2 + \mu\beta_k \tilde{w}_k^\top V_j a_k \right\}, \quad (21)$$

where $A_{j,C} = \{a_k : k \in S_j \cup C\} \in \mathbb{R}^{(n+m) \times n}$ and

$$z_{ijC}(A_{j,C}) = \eta p_j^{-1} \exp(\eta p_j^{-1}) \Bigg( \exp(-\tilde{u}_i) + \sum_{k \in S_j - \{i\}} e^{x_i^\top(\tilde{w}_k - \tilde{w}_{y_i}) - \tilde{u}_i} e^{b_i^\top(a_k - a_{y_i})}$$

$$+ \frac{K - n}{m} \sum_{k \in C} e^{x_i^\top(\tilde{w}_k - \tilde{w}_{y_i}) - \tilde{u}_i} e^{b_i^\top(a_k - a_{y_i})} \Bigg).$$

---

[11]We have assumed here that $dim(span(\{x_i : i \in S_j\})) = n$, which will be most often the case. If the dimension of the span is lower than $n$ then let $V_j$ be of dimension $D \times dim(span(\{x_i : i \in S_j\}))$.

[12]Note that we have used $\tilde{w}_k$ instead of $\tilde{w}_k^{\parallel}$ in writing the parallel component optimization problem. This does not make a difference as $\tilde{w}_k$ always appears as an inner product with a vector in the span of $\{x_i : i \in S_j\}$.

The $e^{b_i^\top(a_k - a_{y_i})}$ factors come from

$$
\begin{aligned}
x_i^\top w_k &= x_i^\top (\tilde{w}_k^\| + a_k^\top V_j) \\
&= x_i^\top \tilde{w}_k + (V_j b_i)^\top V_j a_k \\
&= x_i^\top \tilde{w}_k + b_i^\top V_j^\top V_j a_k \\
&= x_i^\top \tilde{w}_k + b_i^\top a_k,
\end{aligned}
$$

since $V_j$ is an orthonormal basis.

### F.3.3 OPTIMIZING THE IMPLICIT SGD UPDATE EQUATION

To optimize (21) we need to be able to take the derivative:

$$
\begin{aligned}
\nabla_{a_\ell} &\left( \sum_{i \in S_j} (1 + P(z_{ijC}(A_{j,C})))^2 + \sum_{k \in S_j \cup C} (1 + \frac{\mu \beta_k}{2}) \|a_k\|_2^2 + \mu \beta_k \tilde{w}_k^\top V_j a_k \right) \\
&= \sum_{i \in S_j} 2(1 + P(z_{ijC}(A_{j,C}))) \partial_{z_{ijC}(A_{j,C})} P(z_{ijC}(A_{j,C})) \nabla_{a_\ell} z_{ijC}(A_{j,C}) \\
&\qquad + (2 + \mu \beta_\ell) a_\ell + \mu \beta_\ell \tilde{w}_\ell^\top V_j \\
&= \sum_{i \in S_j} 2(1 + P(z_{ijC}(A_{j,C}))) \frac{P(z_{ijC}(A_{j,C}))}{z_{ijC}(A_{j,C})(1 + P(z_{ijC}(A_{j,C})))} \nabla_{a_\ell} z_{ijC}(A_{j,C}) \\
&\qquad + (2 + \mu \beta_\ell) a_\ell + \mu \beta_\ell \tilde{w}_\ell^\top V_j \\
&= \sum_{i \in S_j} 2 \frac{P(z_{ijC}(A_{j,C}))}{z_{ijC}(A_{j,C})} \nabla_{a_\ell} z_{ijC}(A_{j,C}) + (2 + \mu \beta_\ell) a_\ell + \mu \beta_\ell \tilde{w}_\ell^\top V_j \\
&= \sum_{i \in S_j} 2 e^{-P(z_{ijC}(A_{j,C}))} \nabla_{a_\ell} z_{ijC}(A_{j,C}) + (2 + \mu \beta_\ell) a_\ell + \mu \beta_\ell \tilde{w}_\ell^\top V_j
\end{aligned}
$$

where we used that $\partial_z P(z) = \frac{P(z)}{z(1 + P(z))}$ and $e^{-P(z)} = P(z)/z$. To complete the calculation of the derivate we need,

$$
\begin{aligned}
\nabla_{a_\ell} z_{ijC}(A_{j,C}) &= \nabla_{a_\ell} \eta p_j^{-1} \exp(\eta p_j^{-1}) \left( \exp(-\tilde{u}_i) + \sum_{k \in S_j - \{i\}} e^{x_i^\top (\tilde{w}_\ell - \tilde{w}_{y_i}) - \tilde{u}_i} e^{b_i^\top (a_\ell - a_{y_i})} \right. \\
&\qquad \left. + \frac{K-n}{m} \sum_{k \in C} e^{x_i^\top (\tilde{w}_\ell - \tilde{w}_{y_i}) - \tilde{u}_i} e^{b_i^\top (a_\ell - a_{y_i})} \right) \\
&= \eta p_j^{-1} \exp(\eta p_j^{-1}) b_i \\
&\quad \cdot \left( I[\ell \in S_j - \{i\}] e^{x_i^\top (\tilde{w}_\ell - \tilde{w}_{y_i}) - \tilde{u}_i} e^{b_i^\top (a_\ell - a_{y_i})} \right. \\
&\qquad + I[\ell \in C] \frac{K-n}{m} e^{x_i^\top (\tilde{w}_\ell - \tilde{w}_{y_i}) - \tilde{u}_i} e^{b_i^\top (a_\ell - a_{y_i})} \\
&\qquad - I[\ell = y_i] \left( \sum_{k \in S_j - \{i\}} e^{x_i^\top (\tilde{w}_\ell - \tilde{w}_{y_i}) - \tilde{u}_i} e^{b_i^\top (a_\ell - a_{y_i})} \right. \\
&\qquad \left. \left. + \frac{K-n}{m} \sum_{k \in C} e^{x_i^\top (\tilde{w}_\ell - \tilde{w}_{y_i}) - \tilde{u}_i} e^{b_i^\top (a_\ell - a_{y_i})} \right) \right).
\end{aligned}
$$

In order to calculate the full derivate with respect to $A_{j,C}$ we need to calculate $b_i^\top a_k$ for all $i \in S_j$ and $k \in S_j \cup C$. This is a total of $n(n + m)$ inner products of $n$-dimensional vectors, costing $O(n^2(n + m))$. To find the optimum of (21) we can use any optimization procedure that only uses gradients. Since (21) is strongly convex, standard first order methods can solve to $\epsilon$ accuracy in $O(\log(\epsilon^{-1}))$ iterations (Boyd & Vandenberghe, 2004, Sec. 9.3). Thus once we can calculate all of the terms in (21), we can solve it to $\epsilon$ accuracy in runtime $O(n^2(n + m) \log(\epsilon^{-1}))$.

Once we have solved for $A_{j,C}$, we can reconstruct the optimal solution for the parallel component of $w_k$ as $w_k^{\|} = \tilde{w}_k^{\|} + V_j a_k$. Recall that the solution to the perpendicular component is $w_k^{\perp} = \frac{1}{1+\mu\beta_k/2} \tilde{w}_k^{\perp}$. Thus our optimal solution is $w_k = \tilde{w}_k^{\|} + V_j a_k + \frac{1}{1+\mu\beta_k/2} \tilde{w}_k^{\perp}$.

If the features $x_i$ are sparse, then we'd prefer to do a sparse update to $w$, saving computation time. We can achieve this by letting

$$w_k = \gamma_k \cdot r_k$$

where $\gamma_k$ is a scalar and $r_k$ a vector. Updating $w_k = \tilde{w}_k^{\|} + V_j a_k + \frac{1}{1+\mu\beta_k/2} \tilde{w}_k^{\perp}$ is equivalent to

$$\gamma_k = \tilde{\gamma}_k \cdot \frac{1}{1+\mu\beta_k/2}$$

$$r_k = \tilde{r}_k + \mu\beta_k/2 \cdot \tilde{r}_k^{\|} + \tilde{\gamma}_k^{-1}(1 + \mu\beta_k/2) \cdot V_j a_k.$$

Since we only update $r_k$ along the span of $\{x_i : i \in S_j\}$, its update is sparse.

### F.3.4 RUNTIME

There are two major tasks in calculating the terms in (21). The first is to calculate $x_i^{\top} \tilde{w}_k$ for $i \in S_j$ and $k \in S_j \cup C$. There are a total of $n(n + m)$ inner products of $D$-dimensional vectors, costing $O(n(n + m)D)$. The other task is to find the orthonormal basis $V_j$ of $\{x_i : i \in S_j\}$, which can be achieved using the Gram-Schmidt process in $O(n^2 D)$. We'll assume that $\{V_j : j = 1, ..., J\}$ is computed only once as a pre-processing step when defining the mini-batches. It is exactly because calculating $\{V_j : j = 1, ..., J\}$ is expensive that we have fixed mini-batches that do not change during the optimization routine.

Adding the cost of calculating the $x_i^{\top} \tilde{w}_k$ inner products to the costing of optimizing (21) leads to the claim that solve the Implicit SGD update formula to $\epsilon$ accuracy in runtime $O(n(n+m)D + n^2(n + m)\log(\epsilon^{-1})) = O(n(n + m)(D + n\log(\epsilon^{-1})))$.

### F.3.5 INITIALIZING THE IMPLICIT SGD OPTIMIZER

As was the case in Section F.1, it is important to initialize the optimization procedure at a point where the gradient is relatively small and can be computed without numerical issues. These numerical issues arise when an exponent $x_i^{\top} (\tilde{w}_k - \tilde{w}_{y_i}) - \tilde{u}_i + b_i^{\top} (a_k - a_{y_i}) \gg 0$. To ensure that this does not occur for our initial point, we can solve the following linear problem,[13]

$$R = \min_{A_{j,C}} \sum_{k \in C \cup S_j} \|a_k\|_1$$

$$s.t. \quad x_i^{\top} (\tilde{w}_k - \tilde{w}_{y_i}) - \tilde{u}_i + b_i^{\top} (a_k - a_{y_i}) \le 0 \quad \forall i \in S_j, \, k \in C \cup S_j \qquad (22)$$

Note that if $k = y_i$ then the constraint $0 \ge x_i^{\top} (\tilde{w}_k - \tilde{w}_{y_i}) - \tilde{u}_i + b_i^{\top} (a_k - a_{y_i}) = -\tilde{u}_i$ is automatically fulfilled since $\tilde{u}_i \ge 0$. Also observed that setting $a_k = -V_j^{\top} \tilde{w}_k$ satisfies all of the constraints, and so

$$R \le \sum_{k \in C \cup S_j} \|V_j^{\top} \tilde{w}_k\|_1 \le (n + m) \max_{k \in C \cup S_j} \|V_j^{\top} \tilde{w}_k\|_1.$$

We can use the solution to (22) to gives us an upper bound on (21). Consider the optimal value $A_{j,C}^{(R)}$ of the linear program in (22) with the value of the minimum being $R$. Since $A_{j,C}^{(R)}$ satisfies the constrain in (22) we have $z_{ijC}(A_{j,C}^{(R)}) \le K\eta p_j^{-1} \exp(\eta p_j^{-1})$. Since $P(z)$ is a monotonically increasing function that is non-negative for $z \ge 0$ we also have $(1 + P(z_{ijC}(A_{j,C}^{(R)})))^2 \ge (1 + P(K\eta p_j^{-1} \exp(\eta p_j^{-1})))^2$. Turning to the norms, we can use the fact that $\|a\|_2 \le \|a\|_1$ for any

---

[13]Instead bounding the constraints on the right with 0, we could also have used any small positive number, like 5.

vector $a$ to bound

$$\sum_{k \in S_j \cup C} (1 + \frac{\mu \beta_k}{2}) \|a_k\|_2^2 + \mu \beta_k \tilde{w}_k^\top V_j a_k$$

$$\leq \sum_{k \in S_j \cup C} (1 + \frac{\mu \beta_k}{2}) \|a_k\|_1^2 + \mu \beta_k \|\tilde{w}_k^\top V_j\|_1 \|a_k\|_1$$

$$\leq \left( 1 + \mu \cdot \max_{k \in S_j \cup C}\{\beta_k\}/2 \right) \sum_{k \in S_j \cup C} \|a_k\|_1^2 + \mu \max_{k \in S_j \cup C}\{\beta_k \|\tilde{w}_k^\top V_j\|_1\} \sum_{k \in S_j \cup C} \|a_k\|_1$$

$$\leq \left( 1 + \mu \cdot \max_{k \in S_j \cup C}\{\beta_k\}/2 \right) R^2 + \mu \max_{k \in S_j \cup C}\{\beta_k\} \max_{k \in S_j \cup C}\{\|\tilde{w}_k^\top V_j\|_1\} R$$

$$\leq \left( 1 + \mu \cdot \max_{k \in S_j \cup C}\{\beta_k\}/2 \right) \left( (n+m) \max_{k \in C \cup S_j} \|V_j^\top \tilde{w}_k\|_1 \right)^2$$

$$\quad + \mu \max_{k \in S_j \cup C}\{\beta_k\} \max_{k \in S_j \cup C}\{\|\tilde{w}_k^\top V_j\|_1\} \left( (n+m) \max_{k \in C \cup S_j} \|V_j^\top \tilde{w}_k\|_1 \right)$$

$$\leq (1 + \mu \cdot \max_{k \in S_j \cup C}\{\beta_k\})(n+m)^2 \max_{k \in C \cup S_j} \|V_j^\top \tilde{w}_k\|_1^2$$

$$\leq (1 + \mu \cdot \max_{k \in S_j \cup C}\{\beta_k\})(n+m)^2 \max_{k \in C \cup S_j} \|\tilde{w}_k\|_1^2.$$

Putting the bounds together we have that the optimal value of (21) is upper bounded by its value at the solution to (22), which in turn is upper bounded by

$$n(1 + P(K\eta p_j^{-1} \exp(\eta p_j^{-1})))^2 + (1 + \mu \cdot \max_{k \in S_j \cup C}\{\beta_k\})(n+m)^2 \max_{k \in C \cup S_j} \|\tilde{w}_k\|_1^2.$$

This bound is guarantees that our initial iterate will be numerically stable.

## G  LEARNING RATE PREDICTION AND LOSS

Here we present the results of using different learning rates for each algorithm applied to the Eurlex dataset. In addition to the Implicit SGD, NCE, IS, OVE and U-max algorithms, we also provide results for NCE with $n = 1, m = 1$, denoted as NCE (1,1) . NCE and NCE (1,1) have near identical performance.

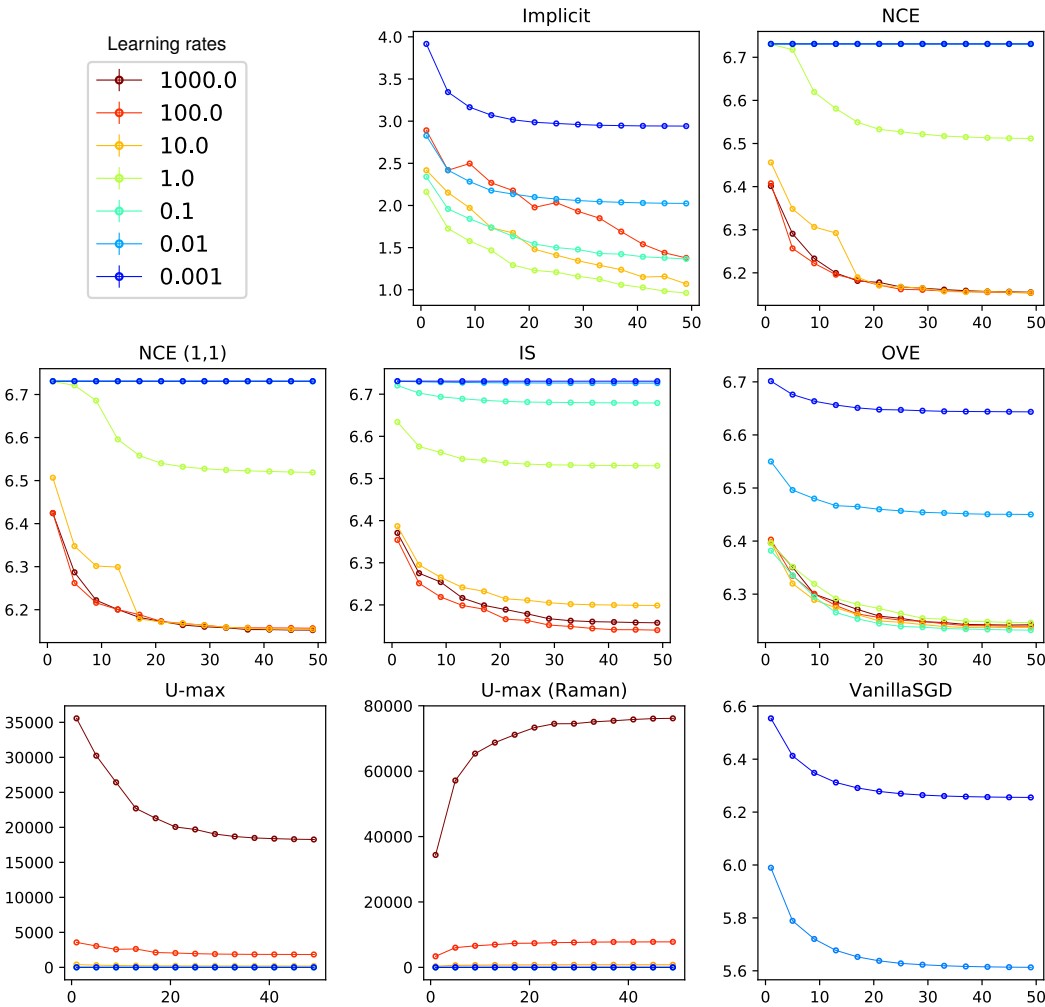

Figure 3: Log-loss on Eurlex different learning rates.

