# OpenReview forum: "Unbiased scalable softmax optimization"
_ICLR.cc/2018/Conference — Reject_

### Official Review · AnonReviewer2 · 2017-11-26
**The paper presents interesting algorithms for minimizing softmax with many classes.**

**Rating:** 5
**Confidence:** 4

**Review:**

The paper presents interesting algorithms for minimizing softmax with many classes. The objective function is a multi-class classification problem (using softmax loss) and with linear model. The main idea is to rewrite the obj as double-sum using the dual formulation and then apply SGD to solve it. At each iteration, SGD samples a subset of training samples and labels. The main contribution of this paper is: 1) proposing a U-max trick to improve the numerical stability and 2) proposing an implicit SGD approach. It seems the implicit SGD approach is better in the experimental comparisons.

I found the paper quite interesting, but meanwhile I have the following comments and questions:

- As pointed out by the authors, the idea of this formulation and doubly SGD is not new. (Raman et al, 2016) has used a similar trick to derive the double-sum formulation and solved it by doubly SGD. The authors claim that  the algorithm in (Raman et al) has an O(NKD) cost for updating u at the end of each epoch. However, since each epoch requires at least O(NKD) time anyway (sometimes larger, as in Proposition 2), is another O(NKD) a significant bottleneck? Also, since the formulation is similar to (Raman et al., 2016), a comparison is needed.

- I'm confused by Proposition 1 and 2. In appendix E.1, the formulation of the update is derived, but why we need Newton to get log(1/epsilon) time complexity? I think most first order methods instead of Newton will have linear converge (log(1/epsilon) time)? Also, I guess we are assuming the obj is strongly convex?

- The step size is selected in one dataset and used for all others. This might lead to divergence of other algorithms, since usually step size depends on data. As we can see, OVE, NCE and IS diverges on Wiki-small, which may be fixed if the step size is chosen for each data (in practice we can choose using subsamples for each data).

- All the comparisons are based on "epochs", but the competing algorithms are quite different and can have very different running time for each epoch. For example, implicit SGD has another iterative solver for each update. Therefore, the timing comparison is needed in this paper to justify that implicit SGD is faster.

- The claim that "implicit SGD never overshoots the optimum" needs more supports. Is it proved in some previous papers?

- The presentation can be improved. I think it will be helpful to state the algorithms explicitly in the main paper.

---

> ### Author Response · Authors · 2017-12-14
> **Comments addressed**
>
> Thank you for your valuable feedback. Below we respond to your points and mention changes that we have made to the paper to address your concerns.
>
> - Our method only requires O(ND) per epoch, since in each epoch we take an O(D) stochastic gradient at each data point. This is a factor of K smaller than Raman et al. In most of our experiments, the second epoch of Raman would not have even started by the time our algorithms had already nearly converged!
>
> The reason why their method is so slow is that they require the denominator (i.e. the partition function) for each data point to be fully computed in each epoch with an associated cost of O(NKD). This is necessary to ensure that their stochastic gradients do not become too large. Our methods never require that the denominator be computed exactly - instead we optimize over u_i which approximates this quantity. As a result our algorithms cost only O(ND), opposed to O(NKD), per epoch.
>
> - The objective in the implicit SGD updates is always strongly convex (this is explicitly stated in the appendix). You are quite correct that first order methods can achieve the log(1/epsilon) rate and that Newton's method is not required (thanks for pointing this out!). This improves the run-time bound of our algorithm. We have adjusted our comments on the bound accordingly.
>
> - The OVE, NCE, IS and U-max algorithms all have virtually the same runtime. Implicit SGD has a longer runtime because it has to solve an optimization problem to compute the update. If multiple data points and multiple classes are sampled each iteration, the run time to compute the update can be significant, and may be more expensive than computing the inner products x^\top w. This will be the case even if first order methods with O(log(1/epsilon)) convergence rates are used. To assess the speed of Implicit SGD in this general setting we agree that a timing comparison would be required.
>
> That said, we have now added a very tight bound for the Implicit SGD update when a single data point and single class is sampled. This is because, in this case, we can update using the bisection method. This is a new result that has been added as Proposition 2. Since the size of the initial bisection interval is provably small, the cost of the bisection search will be less than that of calculating the x^\top w inner products and so Implicit SGD will have very similar runtime to methods like OVE, NCE, IS and U-max. Indeed we found this to be the case in practice.
>
> - Our decision to fix the learning rate using one experiment and then apply this rate to all other datasets was done in order to reflect how the algorithm might be used in practice. However, you suggest a nice alternative and we will follow your suggestion. We will rerun the experiments using 10% of the data to tune the learning rate, and then apply the tuned learning rate to the full dataset. This also gives us the opportunity to compare to vanilla SGD, which was unstable on most datasets except for very small learning rates (which we will now tune for each dataset). We will add the results to the paper once the experiments are complete.
>
> - Our claim that "Implicit SGD never overshoots the optimum" was limited to the quadratic for which we gave the explicit Implicit SGD update formula: \theta^{(t+1)} = \theta^{(t)}/(1+\eta_t). In general it is possible for Implicit SGD to overshoot the optimum. We will improve our wording to clarify this.
>
> - We will work on the text to improve its quality.

---

### Official Review · AnonReviewer1 · 2017-11-27
**Paper explores SGD based explorations to reduce instability in soft-max minimization**

**Rating:** 5
**Confidence:** 3

**Review:**

The problem of numerical instability in applying SGD to soft-max minimization is the motivation. It would have been helpful if the author(s) could have made a formal statement.
Since the main contributions are two algorithms for stable SGD it is not clear how one can formally say that they are stable. For this a formal problem statement is necessary. The discussion around eq (7) is helpful but is intuitive and it is difficult to get a formal problem which we can use later to examine the proposed algorithms.

The proposed algorithms are variants of SGD but it is not clear why they should converge faster than existing strategies.
Some parts of the text are badly written, see for example the following line(see paragraph before Sec 3)

"Since the converge of SGD is
inversely proportional to the magnitude of its gradients (Lacoste-Julien et al., 2012), we expect the
formulation to converge faster."

which could have shed more light on the matter.

The title is also misleading in using the word "exact". I have understand it correct the proposed SGD method solves the optimization problem to an additive error.

In summary the algorithms are novel variants of SGD but the associated claims of numerical stability and speed of convergence vis-a-vis existing methods are missing. The choice of word exact is also not clear.

---

> ### Author Response · Authors · 2017-12-14
> **Comments addressed**
>
> Thank you for your valuable feedback. Below we respond to your points and mention changes that we have made to the paper to address your concerns.
>
> - You are correct to point out that "The problem of numerical instability in applying SGD to soft-max minimization [formulated as double sum] is the motivation". We will make this clearer in the text.
>
> - Our response to the comment that we have not formally proven the stability of our algorithms is as follows. The stability of all these algorithms is essentially determined by the magnitude of the gradient vector. For learning rates that are sufficiently large to have reasonably fast convergence, the magnitude of the gradients should always be small enough so that the step size (i.e. the product of the learning rate and the gradient) does not overflow numerical precision or grossly overshoot the optimum. In the original version, we provided a bound on the magnitude of the gradients for U-max (see the paragraph under Theorem  1). We have now added Proposition 3 that proves that the magnitude of the Implicit SGD step size is O(x_i^\top (w_k - w_{y_i})-u_i) (as opposed to vanilla SGD which has an exponentially larger bound). Thus, we have formal bounds on the magnitude of the stochastic gradients for both of our algorithms.
>
> You correctly observe that the exponential form of the vanilla stochastic gradients was the cause of its instability. Having bounded gradients, as our algorithms do, directly resolves this source of instability. This was confirmed in our (yet to be added - see below) experiments where our algorithms were stable for learning rates where vanilla SGD was not.
>
> - Since Implicit SGD has roughly the same convergence rates as vanilla SGD, we do not make any formal claims of faster convergence for our methods. However, since the gradients of our methods are bounded, we expect their variance to be lower and the allowed step length to be larger, leading to faster convergence in practice. To make these claims testable, we will add empirical comparisons to vanilla SGD where we use sufficiently small learning rates to ensure that it is stable. See our response to the reviewer #2 for more details on our experiment design.
>
> - By "exact" we were implying "converges to the optimal MLE". Although the MLE is an efficient estimator, for a finite number of samples, its optimum will differ from the true parameters. We will change "exact" to "unbiased", which should resolve potential confusion.
>
> - We will work on the text to improve its quality.

---

### Official Review · AnonReviewer3 · 2017-12-13
**Interesting, but with flaws.**

**Rating:** 5
**Confidence:** 4

**Review:**

The paper develops an interesting approach for solving multi-class classification with softmax loss.

The key idea is to reformulate the problem as a convex minimization of a "double-sum" structure via a simple conjugation trick.  SGD is applied to the reformulation: in each step samples a subset of the training samples and labels, which appear both in the double sum.  The main contributions of this paper are: "U-max" idea (for numerical stability reasons) and an ""proposing an "implicit SGD" idea.

Unlike the first review, I see what the term "exact" in the title is supposed to mean. I believe this was explained in the paper. I agree with the second reviewer that the approach is interesting. However, I also agree with the criticism (double sum formulations exist in the literature; comments about experiments); and will not repeat it here. I will stress though that the statement about Newton in the paper is not justified. Newton method does not converge globally with linear rate. Cubic regularisation is needed for global convergence. Local rate is quadratic.

I believe the paper could warrant acceptance if all criticism raised by reviewer 2 is addressed.

I apologise for short and late review: I got access to the paper only after the original review deadline.

---

> ### Author Response · Authors · 2017-12-14
> **Comments addressed**
>
> Thank you for your valuable feedback. Since your comments overlap significantly with reviewer #2, we have addressed your points in the response to reviewer #2.

---

### Author Response · Authors · 2018-01-05
**Modified original submission**

The modified paper with the recommended changes has been uploaded. We believe that the paper has been much improved by addressing the concerns raised by the reviewers (thank you again for your feedback). In particular the new bounds on Implicit SGD's step size and computation using the bisection method give new insights into its performance and make it a more reliable method. The new experiments have very similar results to the old experiments, with Implicit SGD outperforming all other methods.

---

### Decision · Program_Chairs · 2018-01-29
**ICLR 2018 Conference Acceptance Decision**

**Decision:**

Reject

**Comment:**

The key concern from the reviewers that was not addressed is that none of the experimental results illustrate convergence vs. time instead of convergence vs. number of iterations.  While the authors point out that their method is O(ND) instead of O(KND), the reviewers really wanted to see graphs demonstrating this, given that the implicit SGD method requires an iterative solver. The revised paper is otherwise much improved from the original submission, but falls a bit short of ICLR acceptance because of the lack of a measurement of convergence vs. time.

Pros:
+ Promising unbiased algorithms for optimizing the log-likelihood of a model using a softmax without having to repeatedly compute the normalizing factor.

Cons:
- The key concern from the reviewers that was not addressed is that none of the experimental results illustrate convergence vs. time instead of convergence vs. number of iterations.